# Ankyrin-B is a PI3P effector that promotes polarized α5β1-integrin recycling via recruiting RabGAP1L to early endosomes

Fangfei Qu[1,2,3,4], Damaris N Lorenzo[1,2,3,4], Samantha J King[5,6], Rebecca Brooks[5,6], James E Bear[5,6], Vann Bennett[1,2,3,4]*

[1]Department of Biochemistry, Duke University Medical Center, Durham, United States; [2]Department of Cell Biology, Duke University Medical Center, Durham, United States; [3]Department of Neurobiology, Duke University Medical Center, Durham, United States; [4]Howard Hughes Medical Institute, Duke University Medical Center, Durham, United States; [5]UNC Lineberger Comprehensive Cancer Center, The University of North Carolina at Chapel Hill, Durham, United States; [6]Department of Cell Biology and Physiology, The University of North Carolina at Chapel Hill, Chapel Hill, United States

*For correspondence: vann.
bennett@duke.edu

Competing interests: The authors declare that no competing interests exist.

**Abstract** Endosomal membrane trafficking requires coordination between phosphoinositide lipids, Rab GTPases, and microtubule-based motors to dynamically determine endosome identity and promote long-range organelle transport. Here we report that ankyrin-B (AnkB), through integrating all three systems, functions as a critical node in the protein circuitry underlying polarized recycling of α5β1-integrin in mouse embryonic fibroblasts, which enables persistent fibroblast migration along fibronectin gradients. AnkB associates with phosphatidylinositol 3-phosphate (PI3P)-positive organelles in fibroblasts and binds dynactin to promote their long-range motility. We demonstrate that AnkB binds to Rab GTPase Activating Protein 1-Like (RabGAP1L) and recruits it to PI3P-positive organelles, where RabGAP1L inactivates Rab22A, and promotes polarized trafficking to the leading edge of migrating fibroblasts. We further determine that α5β1-integrin depends on an AnkB/RabGAP1L complex for polarized recycling. Our results reveal AnkB as an unexpected key element in coordinating polarized transport of α5β1-integrin and likely of other specialized endocytic cargos.

## Introduction

The currently accepted view that plasma membranes of eukaryotic cells are in a state of flux due to rapid internalization and recycling of membrane proteins received its first experimental support in a prescient 1976 paper by Ralph Steinman and his colleagues (*Steinman et al., 1976*). Steinman (later awarded a Nobel Prize for the discovery of dendritic cells) observed that a surface area equivalent to the entire plasma membrane was internalized every 2 hr in L-cells, which implied a rate of turn-over for the bulk plasma membrane much faster than that of individual membrane proteins known at that time. His non-intuitive conclusion was that the majority of internalized membrane was returned to the cell surface in the form of small vesicles:

"A plausible morphologic mechanism for membrane recycling is that it involves the production of tiny vesicles capable of returning large amounts of surface membrane, with very little content, to the cell surface"

**eLife digest** The membranes that surround animal and other eukaryotic cells are in a state of flux. Small fluid-filled sacs known as vesicles form from the membrane and move into the cell, while other vesicles are returning with cargos of chemicals. These processes allow cells to rapidly adjust the composition of their surfaces for different activities, such as migrating to other parts of the body. Like rock climbers, migrating cells need to hang onto nearby surfaces as they move and so vesicles deliver sticky "adhesion" proteins to the front of migrating cells.

At least three different kinds of molecule are involved in delivering vesicles to a particular end of the cell. Fat molecules known as phosphoinositide lipids act as markers to identify different vesicles, while proteins called GTPases determine which direction the third type of molecules (known as molecular motors) will move the vesicles across the cell. However, it is still not clear how these molecules work together to transport specific cargos to the front of cells during migration.

Now, Qu et al. discover that a protein called ankyrin-B coordinates the directional transport of specific vesicles within migrating cells. Biochemical experiments in cells isolated from mouse embryos show that ankyrin-B recognizes vesicles containing specific phosphoinositide lipids and attaches to them. Ankyrin-B then recruits molecular motors and another protein called RabGAP1L, which regulates the activity of GTPases, to direct the movements of the vesicles. Microscopy experiments reveal that this machinery is essential to transport cell adhesion proteins and other specific cargos to the front of migrating cells.

The next step following on from this work will be to examine how ankyrin-B and RabGAP1L behave in cells that are still inside the body of the animal. Future experiments will identify other cargos that use this machinery to reach the cell membrane and investigate how cells recognize and select these cargos for transport.

Steinman's conjecture of membrane recycling through small vesicles was soon validated with the discovery of receptor-mediated endocytosis and of the return of endosomal receptors to the plasma membrane following separation from ligands (*Pearse and Bretseher, 1981*). Membrane recycling is now recognized as a fundamental property of nucleated cells required for diverse functions, including cell migration, cytokinesis, receptor signaling, and synaptic transmission.

Much progress has been made in elucidating the molecular components required for the highly selective sorting and trafficking of Steinman's 'tiny vesicles', now termed endosomes. These include identification of numerous small GTPases, as well as of phosphoinositide lipids, together with their effectors and regulators, which collaborate to determine the endosomal identity (*Balla, 2013*; *Jean and Kiger, 2012*). In addition, diverse molecular motors that promote both long-range and local endosomal transport have been identified (*Granger et al., 2014*).

Despite these remarkable findings, it is not clear how the individual activities associated with endosomes are integrated to promote a highly regulated and precise delivery of particular membrane proteins to specific cellular locations in polarized cells. One plausible mechanism is through scaffolding proteins capable of simultaneously recruiting and modulating the activity of motor and signaling complexes at endosomal membranes. For instance, the molecular scaffolds JNK interacting proteins 1 and 3 (JIP1 and JIP3) promote axonal transport through binding both protein kinases and the small G protein Arf6 on intracellular membranes. Moreover, JIP1 and JIP3 regulate kinesins directly and dynein indirectly through interaction with the dynactin complex (*Fu and Holzbaur, 2013*). The kinesin motor Kif16B provides another example, which, through direct binding of both PI3P lipids and the small G protein Rab14, promotes the transport of FGFR2-endosomes to the fast-growing ends of microtubules during early embryonic development (*Hoepfner et al., 2005*; *Ueno et al., 2011*).

AnkB is a member of the vertebrate ankyrin family of plasma membrane-organizing proteins that, in contrast to Ankyrin-G (AnkG) and Ankyrin-R (AnkR), associates with intracellular organelles (*He et al., 2013*; *Lorenzo et al., 2014*). AnkB promotes axonal transport and growth through coupling dynactin to organelles containing PI3P lipids (*Lorenzo et al., 2014*). AnkB is broadly expressed, suggesting it may coordinate organelle transport in multiple cellular contexts. Here, we

report that AnkB, through binding to PI3P lipids, dynactin, and the Rab GTPase-regulating protein RabGAP1L, functions as a master integrator of endosomal transport, which promotes polarized trafficking of PI3P-positive endosomes bearing α5β1-integrin to the leading edge of migrating fibroblasts.

## Results

### AnkB promotes long-range transport of PI3P-positive organelles

AnkB promotes fast axonal transport by coupling the motor adaptor dynactin to PI3P-positive organelles in neurons (*Lorenzo et al., 2014*). Therefore, we hypothesized that AnkB also contributes to the long-range transport of PI3P-positive organelles in other cell types. We selected primary cultures of mouse embryonic fibroblasts (MEFs) because these cells express 220 kDa AnkB (*Figure 1A*), and are a standard laboratory model that have been extensively studied with respect to organelle transport. We first examined the dynamics of AnkB-positive organelles by tracking the motility of vesicles expressing 220 kDa AnkB-mCherry in AnkB null ($Ank2^{-/-}$) MEFs isolated from postnatal day 0 (PND0) $Ank2^{-/-}$ mice (*Scotland et al., 1998*). It is important to note that over-expression of 220 kDa AnkB is lethal for MEFs, and it was critical to express 220 kDa AnkB-mCherry in an AnkB null background. We observed that a subset of WT AnkB-mCherry-associated organelles exhibited fast long-range motility, with a net velocity greater than 4 µm/s and a persistence greater than 0.5, which is consistent with long-range, fast microtubule-based transport (*Figure 1B*).

To examine the role of AnkB's association with dynactin in organelle motility in MEFs, we tracked the motion of vesicles expressing the mutant DD1320AA AnkB-mCherry protein, unable to associate with the dynactin complex (*Ayalon et al., 2011*). DD1320AA AnkB-mCherry-positive organelles showed shorter-range motility, reduced net velocity (velocity < 4 µm/s, persistence < 0.5), and completely lacked a population with fast long-range motion (velocity > 4 µm/s, persistence > 0.5), which is otherwise observed in $Ank2^{-/-}$ MEFs expressing WT AnkB-mCherry (*Figure 1B* and *Figure 1—figure supplement 1A*). Hence, these results demonstrate that AnkB provides long-range motility to a subset of organelles by coupling them through dynactin to either dynein or kinesin motors.

AnkB harbors a highly conserved basic pocket within the second ZU5 (ZU5$^C$) domain that specifically binds PI3P lipids and is required for AnkB's association with axonal cargos (*Figure 1C*, yellow surface) (*Wang et al., 2012*; *Lorenzo et al., 2014*). To examine if AnkB also associates with organelles through PI3P lipids in MEFs, we labeled PI3P lipids using the GFP-2xFYVE$^{EEA1}$ domain (*Schink et al., 2013*). Live microscopy revealed that over 70% of WT AnkB-mCherry-positive vesicles detected were PI3P-positive, and around 45% of PI3P-positive organelles were associated with WT AnkB-mCherry (*Figure 1D–F* and *Figure 1—figure supplement 1D*). In sharp contrast, mutant R1194A AnkB-mCherry, which cannot bind PI3P lipids (*Lorenzo et al., 2014*) (*Figure 1C* red circle), failed to localize to PI3P-positive organelles, and was instead diffusely distributed throughout the cytoplasm (*Figure 1D–E*).

We next sought to uncover the identity of AnkB-positive structures in live MEFs by coexpressing WT AnkB-mCherry and organelle markers in $Ank2^{-/-}$ MEFs. Although WT AnkB-mCherry expressed was preferentially localized to Rab5-positive early endosomes, it also exhibited partial overlap with Rab11-positive recycling endosomes and LAMP1-positive lysosomes. Moreover, we detected a restricted association of AnkB-mCherry with puncta at mitochondria ends. In contrast, we observed almost no localization of AnkB-mCherry to either Golgi or ER membranes (*Figure 1F* and *Figure 1—figure supplement 1B,D*). We also examined the association of PI3P lipids with multiple organelles in fixed MEFs using the GFP-2xFYVE$^{EEA1}$ probe and antibodies against endogenous organelle-specific proteins. As expected (*Gillooly et al., 2000*; *Di Paolo and De Camilli, 2006*), PI3P lipids were enriched in endo-lysosomal membranes (*Figure 1—figure supplement 1C,E*), which closely resembled AnkB's subcellular distribution pattern in MEFs (*Figure 1—figure supplement 1B,D*). Collectively, these results demonstrate that AnkB associates with multiple organelles through PI3P lipids and promotes their long-range transport through interaction with the dynactin motor protein adaptor complex.

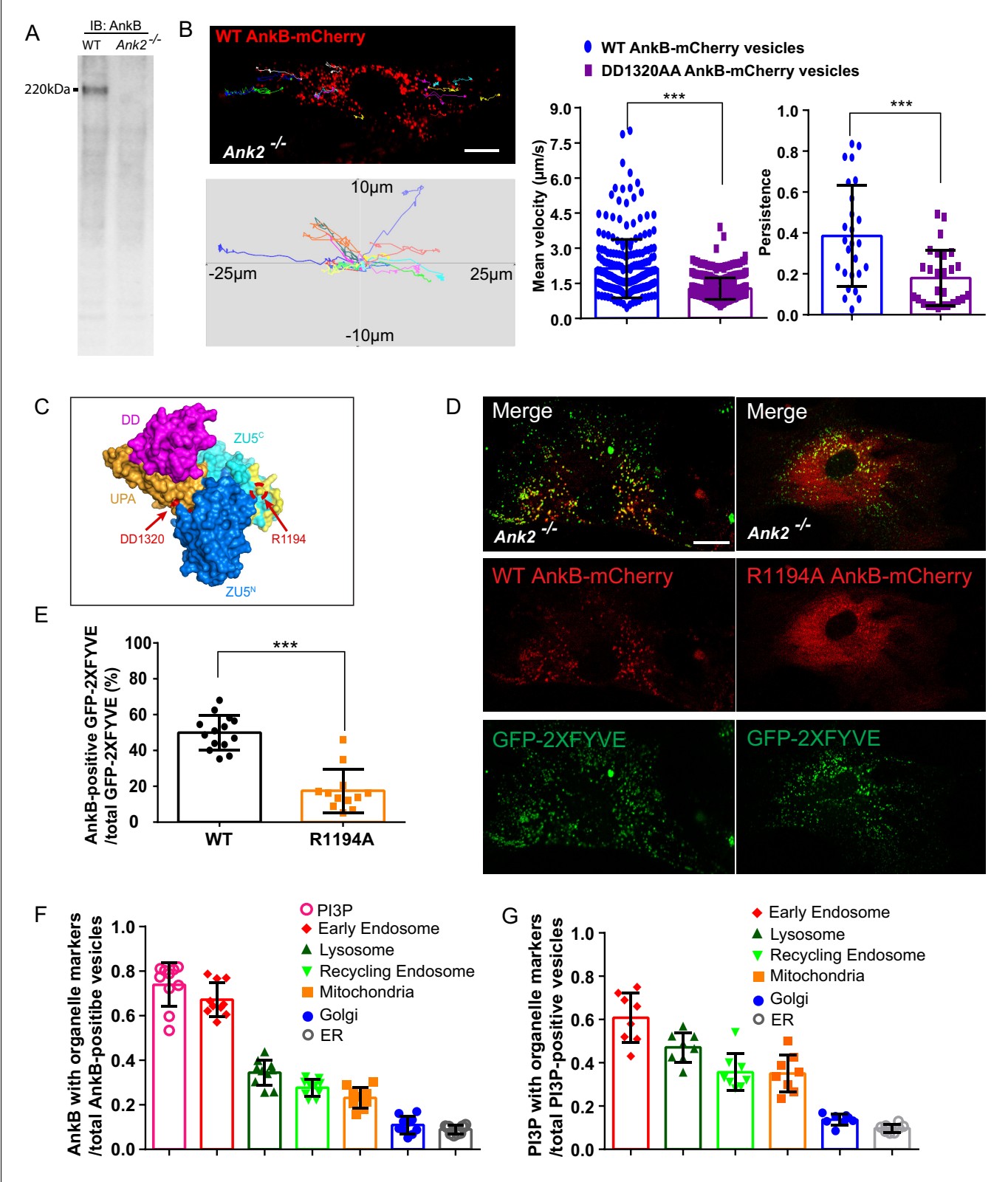

**Figure 1.** AnkB is a PI3P-lipid effector in MEFs. (**A**) AnkB immunoblot (IB) of whole cell lysate from WT and *Ank2*[-/-] MEFs. (**B**) Representative tracks of WT AnkB-mCherry vesicles in *Ank2*[-/-] MEFs and mean velocity and persistence of WT AnkB-mCherry and DD1320AA AnkB-mCherry vesicles. Scale bar, 10 μm. Tracks were plotted in an XY coordinate system assuming (0,0) as initial position. (**C**) Molecular surface representation of the ZU5[N]-ZU5[C]-UPA-DD. The DD1320 site critical for binding to dynactin 4 is pointed by red arrow. Basic residues on the PI3P-binding surface are colored in yellow. The

*Figure 1 continued on next page*

*Figure 1 continued*

R1194 site critical for PI3P binding is circled and pointed in red. (D) Images show the localization of the PI3P biosensor GFP-2×FYVE to WT AnkB-mCherry vesicles in *Ank2*[-/-] MEFs. R1194A AnkB-mCherry was found diffusely distributed in the cytoplasm. Scale bar, 10 μm. (E) Percentage of double mCherry and GFP-positive vesicles. Data in (B) and (E) represent mean ± SD for three independent experiments. ***p<0.001, two-tailed t-test. N = 26 (B), 12 (E). (F–G) Quantitative analysis of localization of WT AnkB-mCherry (F) and GFP-2xFYVE (G) on different organelles. Results are expressed as the ratio of co-localized vesicles over either total AnkB-mCherry or GFP-2xFYVE vesicles. Data represent mean ± SD for three independent experiments. N = 10, 8.

The following figure supplement is available for figure 1:

**Figure supplement 1.** Distribution of AnkB and PI3P lipids on multiple organelles.

## AnkB directly recruits RabGAP1L to PI3P-positive organelles

Rab GTPases regulate endosomal identity and trafficking through the recruitment of effector molecules, and cycle between active (GTP-bound) and inactive (GDP-bound) states, which are respectively determined by GDP/GTP exchange factors (RabGEFs) and GTPase activating proteins (RabGAPs) (*Frasa et al., 2012*). Therefore, we asked whether AnkB interacts with Rab GTPases or their regulators.

AnkB is a multipartite protein with an N-terminus membrane-binding domain (MBD) containing twenty-four ankyrin repeats, a supermodule structure (Zu5[N]-Zu5[C]-UPA domains) that binds β-spectrins, dynactin, and PI3P lipids, a death domain (DD), and an intrinsically unstructured C-terminal regulatory domain, which engages in intramolecular interactions (*Wang et al., 2012*; *Bennett and Lorenzo, 2016*) (*Figure 2A*). Among these domains, only the death domain (named due to structural similarity to death domains of apoptosis-related proteins) has no known partners. Interestingly, a yeast-two-hybrid (Y2H) screen using the death domain of human AnkB (AnkB DD) (*Figure 2A*) and a normalized universal mouse cDNA library identified ten individual positive clones, all sharing the last 65 C-terminal residues of RabGAP1L (*Figure 2B*).

RabGAP1L belongs to the Tre2–Bub2–Cdc16 (TBC) domain-containing family of Rab-specific GTPase-activating proteins (TBC/RabGAPs) (*Fukuda, 2011*), which regulate intracellular membrane trafficking in multiple cellular contexts (*Fuchs et al., 2007*; *Haas et al., 2005*; *Patino-Lopez et al., 2008*). Specifically, RabGAP1L, via a catalytic site on the TBC domain, inactivates Rab22A by promoting its GDP-bound configuration (*Itoh et al., 2006*; *Frasa et al., 2012*). In addition, RabGAP1L contains an N-terminal phosphotyrosine-binding (PTB) domain and a kinesin-like domain of unknown function (*Hidaka et al., 2000*) (*Figure 2B*).

Co-immunoprecipitation (co-IP) and co-localization experiments using affinity-purified antibodies against AnkB and RabGAP1L, which recognize single polypeptides of 220 kDa and 93 kDa in MEFs, respectively (*Figure 1A* and *Figure 2—figure supplement 1A*), showed that endogenous AnkB and RabGAP1L interact in cells (*Figure 2C* top). Interestingly, the AnkG death domain (AnkG DD), which is 65% homologous with the AnkB DD (*Figure 2—figure supplement 1B*), did not interact with RabGAP1L in Y2H assays (data not shown). Similarly, full-length endogenous AnkG did not co-immunoprecipitate with RabGAP1L from MEF lysates (*Figure 2C* bottom). Thus, AnkB either gained the ability to bind to RabGAP1L after the divergence of AnkB and AnkG in early vertebrate evolution, or AnkG lost this activity.

We performed a proximity ligation assay (PLA) as well as immunofluorescence assay to further assess the association of endogenous AnkB and RabGAP1L, and to identify the sites of their interaction in MEFs. PLA produces a positive signal when putative binding partners are within less than 40 nm of each other, which allows the DNA labels of antibodies against these proteins to form double strands (*Söderberg et al., 2006*). Consistent with the Y2H and co-IP results, we observed strong PLA labeling of cytoplasmic organelles in WT MEFs and complete loss of signal in *Ank2*[-/-] MEFs (*Figure 2D*). Immunofluorescent staining of endogenous AnkB and RabGAP1L also re-confirmed their co-localization on a subset of cytoplasmic organelles (*Figure 2—figure supplement 1H*).

We next sought to identify AnkB and RabGAP1L residues critical for their interaction, which could also serve as critical controls in cellular assays. The lack of association between RabGAP1L and AnkG suggested that divergent sites in the sequences of AnkB DD and AnkG DD may mediate the AnkB DD-RabGAP1L interaction. Binding assays between RabGAP1L and AnkB with alanine mutations in

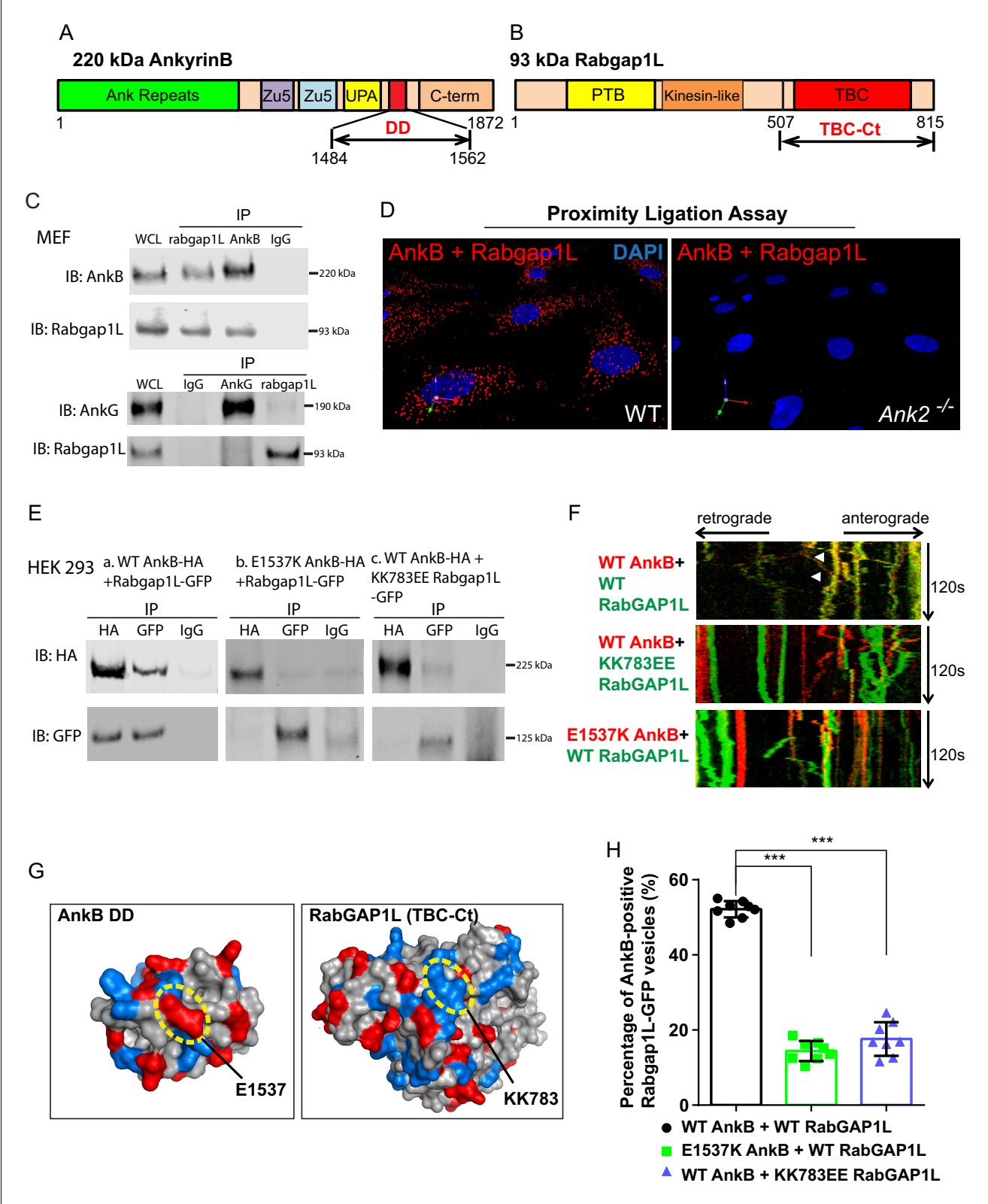

**Figure 2.** RabGAP1L binds to the death domain of AnkB. (A, B) Schematic representation of the domain organization of 220 kDa AnkB (A) and 93 kDa RabGAP1L (B). (C) Co-immunoprecipitation (Co-IP) of endogenous AnkB and RabGAP1L (top), AnkG and RabGAP1L (bottom) from WT MEF lysates. (D) Proximity ligation assay. Red dots indicate cellular sites of interaction between AnkB and RabGAP1L, blue shows nuclear staining. (E) Co-IP of WT AnkB-HA and RabGAP1L-GFP (left), E1537K AnkB-HA and RabGAP1L-GFP (middle), and WT AnkB-HA and KK783EE RabGAP1L-GFP (right) from

*Figure 2 continued on next page*

*Figure 2 continued*

HEK293 cells expressing corresponding plasmids. (**F**) Kymographs of WT AnkB-mCherry and RabGAP1L-GFP (top), WT AnkB-mCherry and KK783EE RabGAP1L-GFP (middle), and E1537K AnkB-mCherry and WT RabGAP1L-GFP (bottom) motion in *Ank2*$^{-/-}$ MEFs. White arrowheads indicate vesicles showing AnkB-mCherry and RabGAP1L co-transport. (**G**) Molecular surface representation of AnkB death domain (AnkB DD) and RabGAP1L TBC-C-terminal domain (TBC-Ct). Residues critical for interaction are highlighted by yellow circles. (**H**) Co-localization of either WT AnkB-mCherry with WT RabGAP1L-GFP, E1537K AnkB-mCherry with WT RabGAP1L-GFP, or WT AnkB-mCherry with KK783EE RabGAP1L-GFP in *Ank2*$^{-/-}$ MEFs. Data represent mean ± SD from three independent experiments. ***p<0.001, one-way ANOVA with Tukey post-test. N = 8.

The following figure supplement is available for figure 2:

**Figure supplement 1.** Validation of the RabGAP1L antibody and AnkB/RabGAP1L interaction.

seven of its DD charged residues diverging from AnkG's DD (*Figure 2—figure supplement 1B*, red box) showed that the E1537A substitution significantly weakened the interaction with RabGAP1L, while the reverse-charge mutant E1537K blocked their association (*Figure 2—figure supplement 1C*). The E1537 site, highly conserved among vertebrate AnkB proteins, resides on the surface of the crystal structure of AnkB DD (*Figure 2G* and *Figure 2—figure supplement 1F*).

Within the AnkB DD interacting portion of RabGAP1L, the highly conserved, positively charged residues 783KKLKK (*Figure 2—figure supplement 1D*) provided potential candidates for interaction with AnkB DD. We found that reversing the charge of the surface exposed 783KK residues to EE (*Figure 2G*) abolishes RabGAP1L binding to AnkB DD in Y2H assays (data not shown). Co-immuno-precipitation from HEK293 cell lysates of full length WT, but not E1537K, AnkB-HA with WT Rab-GAP1L-GFP; as well as of WT, but not KK783EE, RabGAP1L-GFP with WT AnkB-HA (*Figure 2E* and *Figure 2—figure supplement 1E*) corroborated these results.

Time-lapse video microscopy assessing the dynamic localization of AnkB-mCherry and Rab-GAP1L-GFP co-expressed in *Ank2*$^{-/-}$ MEFs revealed that AnkB co-transports with RabGAP1L. Kymograph analysis confirmed that both proteins were co-localized and co-transported on motile vesicles (*Figure 2F* top, 2 hr and *Figure 2—figure supplement 1G*). In contrast, E1537K AnkB-mCherry, which does not bind RabGAP1L, still localizes to vesicles, but no longer co-transports with Rab-GAP1L-GFP (*Figure 2F* bottom, 2 hr and *Figure 2—figure supplement 1G*). Furthermore, the KK783EE RabGAP1L-GFP mutant that abrogated interaction with AnkB also eliminated RabGAP1L-GFP and AnkB-mCherry vesicular co-localization and co-transport (*Figure 2F* center, 2 hr and *Figure 2—figure supplement 1G*).

We next asked whether RabGAP1L localizes to PI3P-positive organelles in an AnkB-dependent manner. While over 60% of RabGAP1L-mCherry vesicles detected in WT MEFs were associated with PI3P-positive vesicles, strikingly, less than 20% of RabGAP1L-mCherry localized to PI3P-positive compartments in *Ank2*$^{-/-}$MEFs (*Figure 3A,D*). This result is further confirmed by immunofluorescence detection of endogenous RabGAP1L in WT and *Ank2*$^{-/-}$MEFs expressing PI3P indicator, GFP-2xFY-VE$^{EEA1}$ (*Figure 3—figure supplement 1A–B*). Together, these results reveal a new protein-protein interaction between AnkB and RabGAP1L that recruits RabGAP1L to PI3P-positive organelles.

## AnkB promotes dissociation of Rab22A from PI3P-associated organelles through recruitment of RabGAP1L

RabGAP1L preferentially activates the GTPase activity of Rab22A (*Itoh et al., 2006*). Rab22A is a vertebrate Rab GTPase closely related to Rab5 that localizes to early endosomes, where it interacts with the early endosomal antigen 1 (EEA1) and the Rab5 guanine nucleotide exchange factor Rabex-5 (*Kauppi et al., 2002*; *Zhu et al., 2009*). Interestingly, Rab22A facilitates the formation of a specialized subset of early endosomes, implicated in endocytosis as well as recycling of both clathrin-dependent and clathrin-independent cargos (*Holloway et al., 2013*; *Maldonado-Báez and Donald-son, 2013*; *Weigert et al., 2004*). We evaluated whether RabGAP1L recruitment to Rab22A-positive organelles depends on AnkB. While in WT MEFs 40% to 60% of GFP-Rab22A-positive vesicles co-localized with RabGAP1L-mCherry, this co-localization was reduced to less than 15% in *Ank2*$^{-/-}$ MEFs (*Figure 3B,E*). This result is further confirmed by immunofluorescence of endogenous RabGAP1L in WT and *Ank2*$^{-/-}$MEFs expressing GFP-Rab22A (*Figure 3—figure supplement 1C–D*).

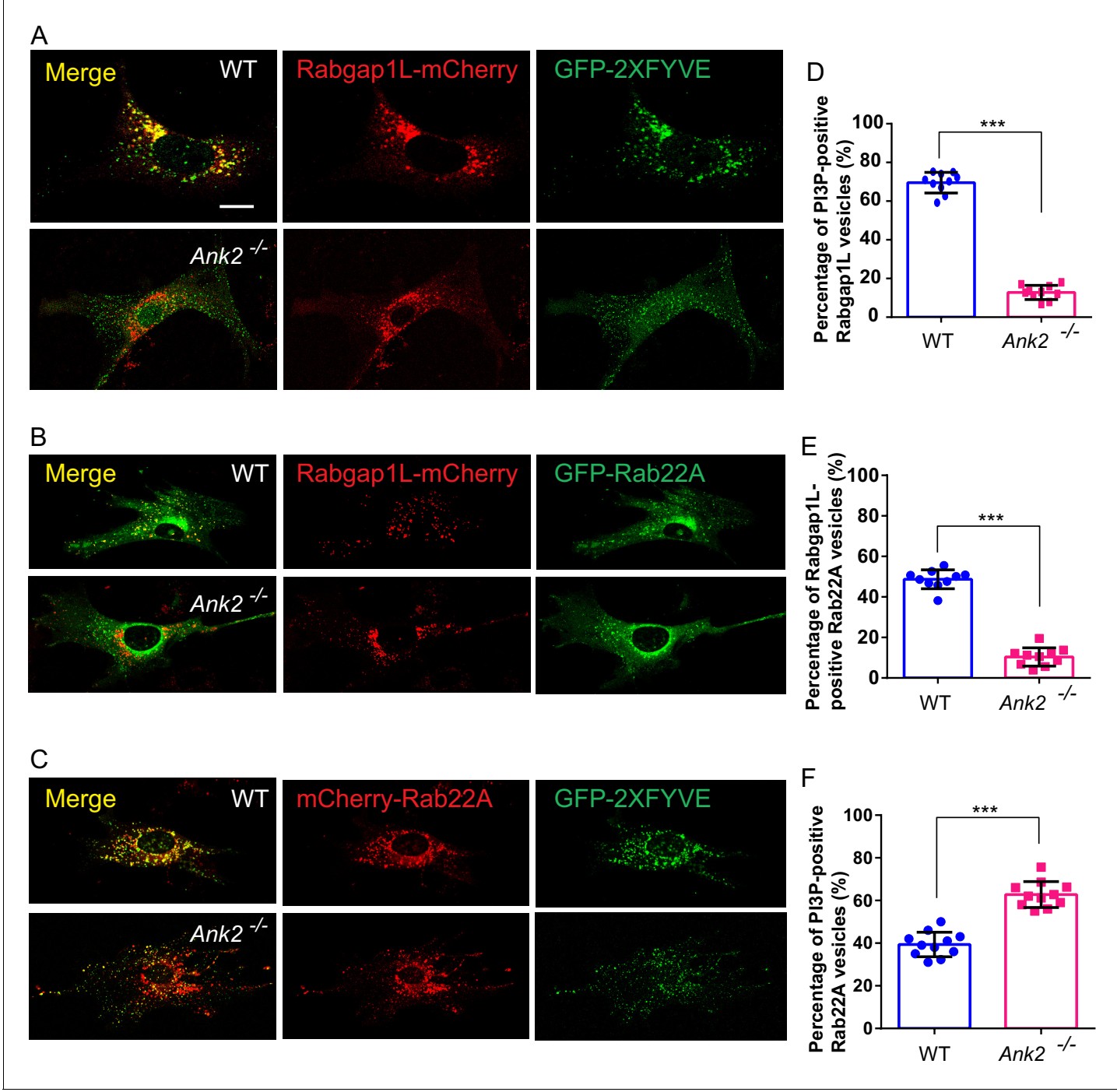

**Figure 3.** AnkB promotes RabGAP1L targeting to Rab22A/PI3P-positive compartments and Rab22A dissociation from PI3P-positive endosomes. (A–C) Representative images of live WT and *Ank2*[-/-] MEFs expressing either RabGAP1L-mCherry and GFP-2xFYVE (A), RabGAP1L-mCherry and GFP-Rab22A (B), or mCherry-Rab22A and GFP-2xFYVE (C). Scale bar, 10 µm. (D) Quantitative data of the percentage of RabGAP1L that localize to GFP-2xFYVE labeled PI3P-positive vesicles in WT and *Ank2*[-/-] MEFs. (E) Quantitative data of the percentage of RabGAP1L that localize to GFP-Rab22A-positive compartments in WT and *Ank2*[-/-] MEFs. (F) Quantitative data of the percentage of Rab22A that localize to GFP-2xFYVE labeled PI3P-positive compartments in WT and *Ank2*[-/-] MEFs. Data represent mean ± SD for three independent experiments. ***$p<0.001$, two-tailed t-test. N = 10, 11.
The following figure supplement is available for figure 3:

**Figure supplement 1.** Distribution of endogenous RabGAP1L to PI3P-positive and Rab22A-positive compartment.

Inactivation of membrane-associated Rab GTPases ensures their dissociation from bound vesicles, which is critical for the transition between endosomal compartments during endosomal trafficking (*Frasa et al., 2012*). Therefore, we speculated that AnkB, via the recruitment of RabGAP1L to PI3P-positive organelles, might promote inactivation of Rab22A and its dissociation from early endosomes. In line with previous reports of Rab22A association with early endosomes (*Kauppi et al., 2002*; *Zhu et al., 2009*), we found that in WT MEFs over 40% of Rab22A localized to PI3P-enriched membranes (*Figure 3C,F*). Remarkably, loss of AnkB expression not only abrogated co-localization of RabGAP1L to Rab22A-positive organelles (*Figure 3B,E*), but also resulted in an increased association of Rab22A with PI3P-positive compartments, especially within the perinuclear region (*Figure 3C,F*). Thus, our data indicate that AnkB promotes dissociation of Rab22A from PI3P-enriched organelles through the recruitment of RabGAP1L.

## AnkB-associated organelles exhibit RabGAP1L-dependent polarized transport

The interaction between RabGAP1L and AnkB prompted us to examine whether AnkB has a role in determining organelle transport polarity. To study the dynamics of AnkB-associated organelles in migrating MEFs, which are polarized cells with well-defined front and rear ends, we tracked their motion from the perinuclear region by expressing AnkB proteins tagged with mMaple3, a photoconvertible molecule that switches from green to red fluorescent emission following blue light exposure (*Wang et al., 2014a*). $Ank2^{-/-}$ MEFs expressing either mMaple3-tagged WT or mutant E1537K AnkB were plated on tissue culture wells containing an insert, which was later removed to allow cell migration into the exposed region (*Figure 4A*). The perinuclear region of migrating cells was pulsed with blue light resulting in conversion of about 70% of green AnkB-mMaple3 to red fluorescent signal (*Figure 4B*, red inset). To dynamically track AnkB-mMaple3 particles, we monitored loss of red fluorescence, due to outward migration of photoconverted perinuclear vesicles, as well as gain of green fluorescence, due to entry of non-photoconverted inward-moving vesicles into the perinuclear region (*Figure 4B–E*).

We found that the rate of entry of green vesicles from peripheral areas was equivalent for both WT and E1537K AnkB (*Figure 4C,E and G*, green symbols). However, photoconverted red WT AnkB vesicles exited the perinuclear region at twice the rate of mutant E1537K AnkB (*Figure 4C,H–I*, red symbols). Moreover, photoconverted perinuclear WT AnkB-mMaple3 red vesicles exhibited a biased transport toward the migrating front of the cell (*Figure 4J* and *Figure 4—figure supplement 1A*). In contrast, red-converted E1537K AnkB-mMaple3 moved from the perinuclear region into both the front and the rear of cells at equivalent rates (*Figure 4J* and *Figure 4—figure supplement 1B*). Taken together, these results demonstrate that AnkB, via interaction with RabGAP1L, coordinates the polarized transport of perinuclear PI3P-positive organelles to the migrating front of MEFs.

## Identification of α5β1-integrin as an AnkB/RabGAP1L-dependent cargo

We next sought to identify membrane protein(s) that are transported via the AnkB/RabGAP1L/Rab22A-mediated pathway in MEFs. Using a cell surface protein biotinylation assay (*Bitsikas et al., 2014*), we found that AnkB deficiency did not cause global changes in either plasma membrane protein internalization or recycling, as shown by the similar rate and extent of internalization and recycling of biotinylated surface proteins to the plasma membrane in WT and $Ank2^{-/-}$ MEFs (*Figure 4—figure supplement 2A–B*). We also found no difference between WT and $Ank2^{-/-}$ MEFs in the dynamics of internalization and recycling of known specialized endocytic cargos, including transferrin (Trn) and αvβ3-integrin (*Figure 4—figure supplement 2C–F*).

We next evaluated the role of AnkB in active transport of the fibronectin receptor α5β1-integrin, which, similar to AnkB, exhibits polarized delivery from cytoplasmic compartments to the leading edge of migrating fibroblasts (*Bretscher, 1989*, *1992*). Moreover, polarized transport of the fibronectin receptor is extensively regulated by GTPases and their adaptor proteins (*Caswell et al., 2008*, *2009*; *Thapa et al., 2012*). We tagged α5-integrin with mMaple3 and tracked its dynamics in migrating WT and $Ank2^{-/-}$ MEFs. In WT MEFs, photoconverted perinuclear α5-integrin-mMaple3 localized to vesicles that were predominantly transported towards the migrating cell front (*Figure 5A–B and G–I* and *Figure 5—figure supplement 1A*). In contrast, in $Ank2^{-/-}$ MEFs, significantly fewer photoconverted perinuclear α5-integrin-mMaple3 exited the converted region, and

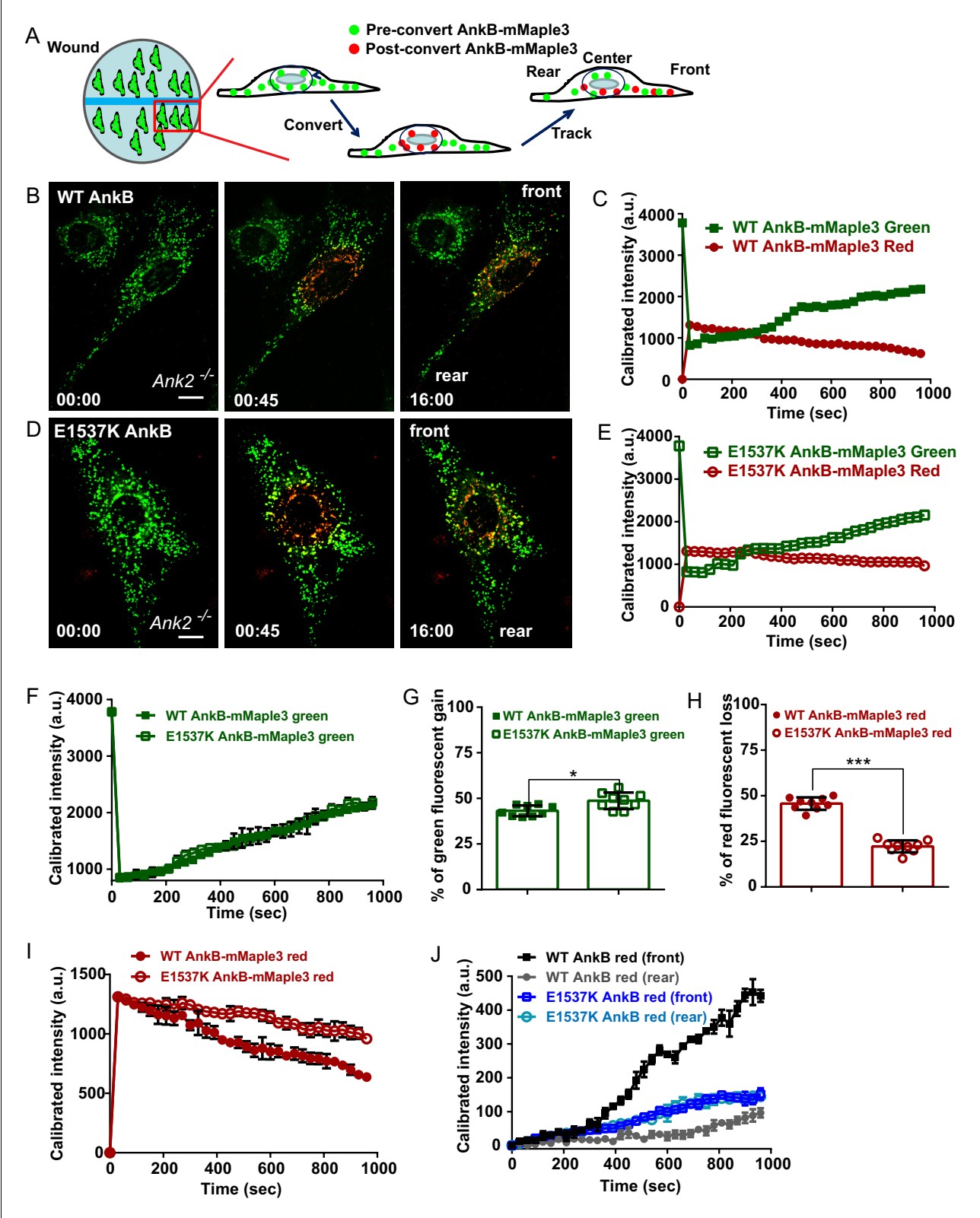

**Figure 4.** AnkB-associated perinuclear endosomal organelles exhibit polarized transport in migrating MEFs. (**A**) Schematic of the experimental design used in the combined photoconversion- wound-induced migration assay. Perinuclear AnkB-mMaple3 signal in *Ank2⁻/⁻* MEF that is migrating at the edge of the wound was converted from green to red fluorescence by blue light exposure. Green or red fluorescent signal were tracked for 16 min following photoconversion. For image representation and analysis cells were divided in front and rear halves determined based on their migratory direction. (**B**
*Figure 4 continued on next page*

*Figure 4 continued*

and D) Representative image of *Ank2⁻/⁻* MEF expressing WT AnkB-mMaple3 (**B**) or E1537K AnkB-mMaple3(**D**) at t = 0 s (pre-converted), 45 s (perinuclear region converted), and 16 min (tracking end point). Scale bar, 10 µm. (**C** and **E**) Quantification of red fluorescent intensity (post-converted AnkB-mMaple3) and green fluorescent intensity (pre-converted AnkB-mMaple3) in the perinuclear region. (**F, I**) Comparative analysis of green fluorescent gain (**F**) and red fluorescent loss (**I**) in the perinuclear region. (**G, H**) Percentage of red fluorescent loss and green fluorescent gain in perinuclear region at 16 min. (**J**) Quantification of red fluorescent intensity gain at either the front or rear cell ends. Intensities at each time point are normalized to the background intensity at t = 0 s. Data represent mean ± SD for three independent experiments. *p=0.011, ***p<0.001, two-tailed t-test. N = 9.

The following figure supplements are available for figure 4:

**Figure supplement 1.** Distribution of WT AnkB-mMaple3 and E1537K AnkB-mMaple3 in MEFs during a photoconversion assay.

**Figure supplement 2.** Global recycling of cell surface proteins, β3-integrin, and transferrin is not affected in *Ank2⁻/⁻* MEFs.

showed no preference in transport directionality (*Figure 5C–D and G–I* and *Figure 5—figure supplement 1B*). Thus, these results indicate that AnkB is required for the polarized transport of α5-integrin from the perinuclear region to the migrating front of MEFs.

The fact that α5β1-integrin has a low degradation rate and is actively recycled in a long-loop microtubule dependent route led us hypothesize that a majority of photo-converted perinuclear α5-integrins belong to the recycling population (*Caswell et al., 2009*; *Bridgewater et al., 2012*; *Arjonen et al., 2012*; *Böttcher et al., 2012*). Therefore, to determine whether α5β1-integrin requires AnkB for polarized recycling, we followed the itinerary of internalized β1-integrin in WT and *Ank2⁻/⁻* MEFs by calculating the percentage of recycled β1-integrin at the cell surface at selected post-internalization times (*Figure 6A*). After 60 min, WT MEFs recycled over 60% of internalized β1-integrins to the plasma membrane (*Figure 6B,D*). In contrast, in *Ank2⁻/⁻* MEFs less than 20% of internalized β1-integrins recycled to the cell surface over the same period, and instead accumulated at the perinuclear region (*Figure 6B,D*). The pool of labeled intracellular β1-integrins in *Ank2⁻/⁻* MEFs eventually did return to the plasma membrane after 3–6 hr (data not shown), with a significantly slower recycling rate than in WT MEFs.

Integration of phosphoinositide lipids, GTPases, and motors is essential for efficient recycling of endocytic cargoes, including α5β1-integrin (*Jović et al., 2007*, *2009*; *Thapa et al., 2012*). Therefore, we addressed the requirement for AnkB interactions with PI3P lipids, dynactin, and RabGAP1L for α5β1-integrin recycling using a structure-function rescue approach. Expression of WT AnkB-mCherry fully restored the rate and extent of β1-integrin recycling to the leading edge of migrating *Ank2⁻/⁻* MEFs (*Figure 6B–D*). In marked contrast, neither E1537K AnkB-mCherry (lacking RabGAP1L binding) nor R1194A AnkB-mCherry (lacking PI3P binding) rescued β1-integrin recycling deficits in *Ank2⁻/⁻* MEFs (*Figure 6C,D* and *Figure 6—figure supplement 1A*). DD1320AA AnkB-mCherry (unable to bind dynactin) (*Figure 6C*) partially rescued β1-integrin recycling to about 50% of the WT levels (*Figure 6D* and *Figure 6—figure supplement 1A*). These results indicate that AnkB's binding to PI3P lipids and RabGAP1L are essential steps in β1-integrin recycling, while its association with the dynactin complex increases its efficiency.

AnkB's MBD is comprised of 24 ANK repeats folded as a solenoid with a peptide-binding groove that mediates binding to multiple membrane proteins (*Bennett and Lorenzo, 2016*). Thus, we hypothesized that AnkB's MBD facilitates β1-integrin recycling via direct interaction with α5β1-integrin or other associated adaptor proteins. To our surprise, expression of a truncated AnkB-mCherry construct lacking the MBD (Zu5-Ct AnkB-mCherry) was sufficient to restore β1-integrin recycling to the leading edge of *Ank2⁻/⁻* MEFs (*Figure 6C,D* and *Figure 6—figure supplement 1B*). Based on these findings, we concluded that integrin recognition through ANK repeats is not required for AnkB-dependent β1-integrin recycling.

Lastly, we investigated whether AnkB is required for either the initial sorting of β1-integrin to early endosomes or for subsequent endosomal maturation steps. The localization of internalized β1-integrin to Rab22A- or Rab5-positive early endosomes, both in WT and *Ank2⁻/⁻* MEFs (*Figure 6—figure supplement 1C–E*), suggests that the initial sorting to early endosomes is independent of AnkB. However, β1-integrin exhibited increased co-localization with Rab5-positive early endosomes and with Rab22A in *Ank2⁻/⁻* MEFs following initiation of recycling (*Figure 6E,F* and *Figure 6—figure*

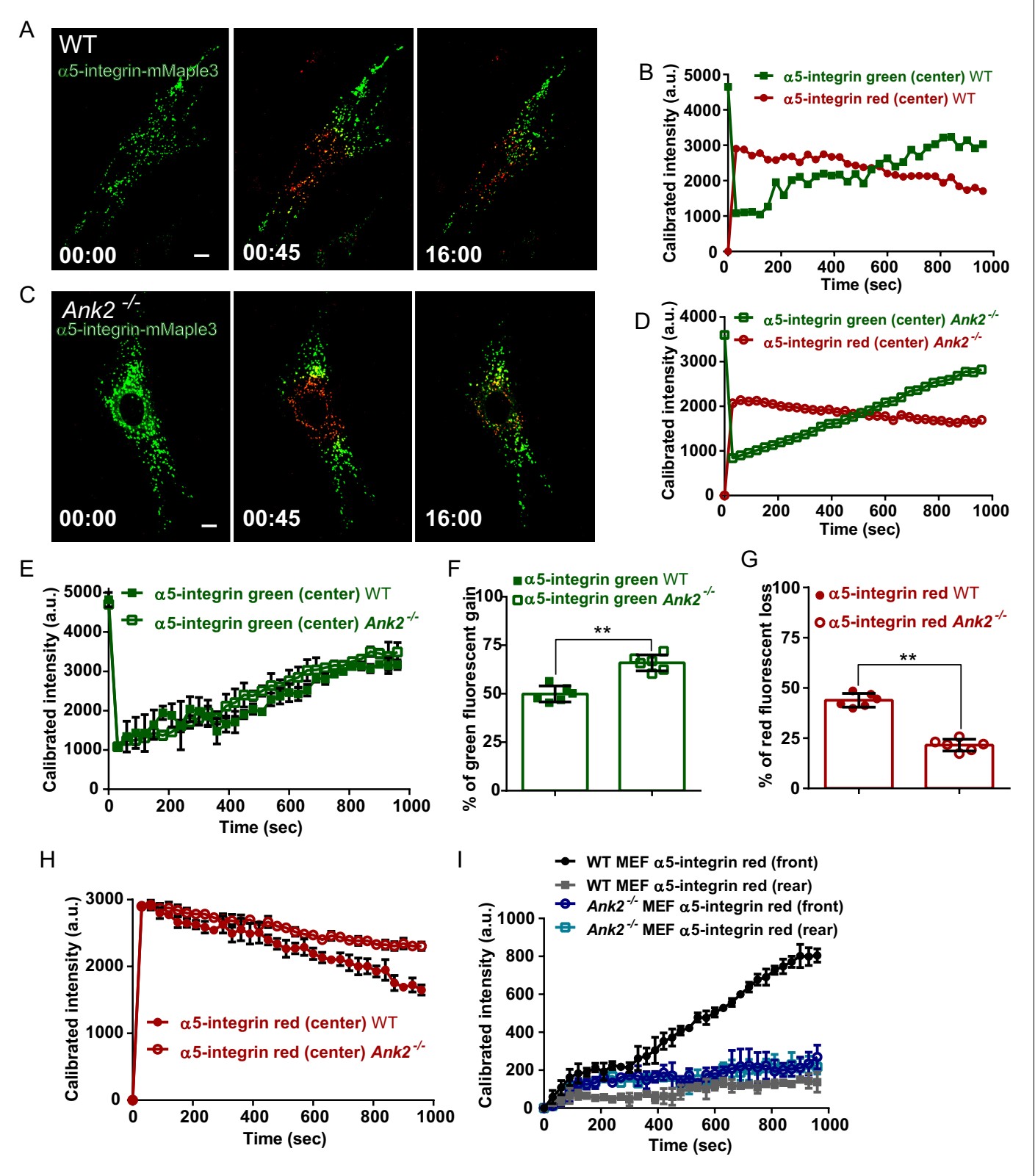

**Figure 5.** AnkB is required for polarized transport of α5-integrin towards the front end of migrating MEFs. (**A** and **C**) Same photoconversion- wound-induced migration assay was performed in WT and *Ank2⁻/⁻* MEFs expressing α5-integrin-mMaple3. Representative images of WT MEFs (**A**) and *Ank2⁻/⁻* MEFs (**C**) expressing α5-integrin-mMaple3 at 0s (pre-converted), 45 s (perinuclear region converted) and16 mins (tracking end point). Scale bar, 10 μm. (**B** and **D**) Quantification of red fluorescent intensity (post-converted α5-integrin-mMaple3) and green fluorescent intensity (pre-converted AnkB-

*Figure 5 continued on next page*

*Figure 5 continued*

mMaple3) in the perinuclear region of WT (**B**) and an *Ank2⁻/⁻* (**D**) MEFs. (**E, H**) Comparative analysis of green fluorescent gain (**E**) and red fluorescent loss (**H**) in the perinuclear region. (**F, G**) Quantitative data of the percentage of red fluorescent loss and green fluorescent gain in the perinuclear region of WT and *Ank2⁻/⁻* MEFs at 16 min. (**I**) Quantification of red fluorescent intensity gain at either the front or rear cell ends. Intensities at each time point are normalized to the background intensity at t = 0 s. Data represent mean ± SD for six independent experiments. Mean from each of the six experiment is shown in **E** and **H**. **p=0.022, two-tailed t-test. N = 13.

The following figure supplement is available for figure 5:

**Figure supplement 1.** Distribution of α5-integrin-mMaple3 in WT and *Ank2⁻/⁻* MEFs during a photoconversion assay.

*supplement 1D*). These results indicate that AnkB, through recruitment of RabGAP1L, facilitates the dissociation of Rab22A from β1-integrin-containing endocytic vesicles to allow their transition from Rab5-positive early endosomes to Rab5-negative recycling endosomal compartments.

## Balanced Rab22A activity is critical for β1-integrin recycling

Spatio-temporal regulation of Rab and Arf GTPase activities is critical for controlling the endosomal recycling of α5β1-integrins (*Caswell et al., 2008*; *Pellinen et al., 2006*; *Li J et al., 2007*). To directly address the requirement of GAP activity of RabGAP1L in β1-integrin recycling, we generated shRNA mediated RabGAP1L knock-down cells. However, cells with knock-down of RabGAP1L or replacement with GAP-deficient RabGAP1L, the R584A mutant that abolish the IxxDxxR arginine finger motif (*Pan et al., 2006*; *Frasa et al., 2012*), exhibited altered morphology and global defects in cell growth (*Figure 6—figure supplement 2*). These results suggest that RabGAP1L also plays important roles though its GAP activity in cellular events other than integrin trafficking and cell migration. Rab21 provides a similar example and is required for β1-integrin trafficking as well as cell adhesion and cytokinesis (*Pellinen et al., 2006*, *2008*; *Mai et al., 2011*).

To test if Rab22A activity is responsible for α5β1-integrin recycling downstream of the AnkB-RabGAP1L pathway, we performed β1-integrin recycling assay in WT MEFs over-expressing either mCherry-tagged WT Rab22A or constitutively active mutant Q64L Rab22A. Interestingly, WT MEFs over-expressing WT Rab22A showed a reduced rate of β1-integrin recycling. The deficiency was further increased in MEFs expressing Q64L Rab22A (*Figure 6—figure supplement 3A–B*). Moreover, we noticed that WT MEFs expressing Q64L Rab22A not only affected internalization of β1-integrins, but also of transferrin and β3-integrins, whose recycling is not affected in the loss of ankB (*Figure 6—figure supplement 3C*). These results suggest that hyper-activation of Rab22A disrupts general recycling of multiple receptors while the AnkB-RabGAP1L mediated regulation specifically tunes the Rab22A activity on selected PI3P-positive organelles bearing β1-integrin.

## AnkB/RabGAP1L interaction promotes directional cell migration

Fibroblasts rely on the efficient recycling of α5β1-integrin adhesion receptors to the plasma membrane for directional migration (*Caswell et al., 2008*; *Mai et al., 2011*; *Jović et al., 2007*; *Zech et al., 2011*), and β1-integrin is required for haptotaxis along fibronectin gradients (*De Franceschi et al., 2015*; *King et al., 2016*). Considering that AnkB promotes α5β1-integrin recycling to the leading edge of migrating MEFs, we next investigated the possibility that AnkB loss impairs directional migration based on a linear fibronectin gradient. Using a microfluidic chamber system-based haptotaxis assay, we tracked the position of individual WT and *Ank2⁻/⁻* MEFs migrating on a linear gradient of fibronectin during a 24 hr interval, which allowed us to calculate the forward migration index (FMI), overall velocity, and persistence of motion (*Wu et al., 2012*) (*Figure 7A*). Typically, fibroblasts haptotaxing on fibronectin gradients display an average FMI above 0.1, with a 95% confidence interval (CI) above 0. In contrast, a FMI of 0 with 95% CI crossing 0 is considered as not haptotaxing (*Wu et al., 2012*). While WT MEFs, similar to control IA32 fibroblasts, exhibited haptotaxis towards higher concentrations of fibronectin, *Ank2⁻/⁻* MEFs migrated in a random pattern (*Figure 7B,C*). Interestingly, *Ank2⁻/⁻* MEFs exhibited no difference in the overall velocity or persistence of the cell migration, suggesting that the cell motility machinery is fully functional in the absence of AnkB (*Figure 7D,E*). The finding that AnkB is required for recycling of α5β1-integrin, but

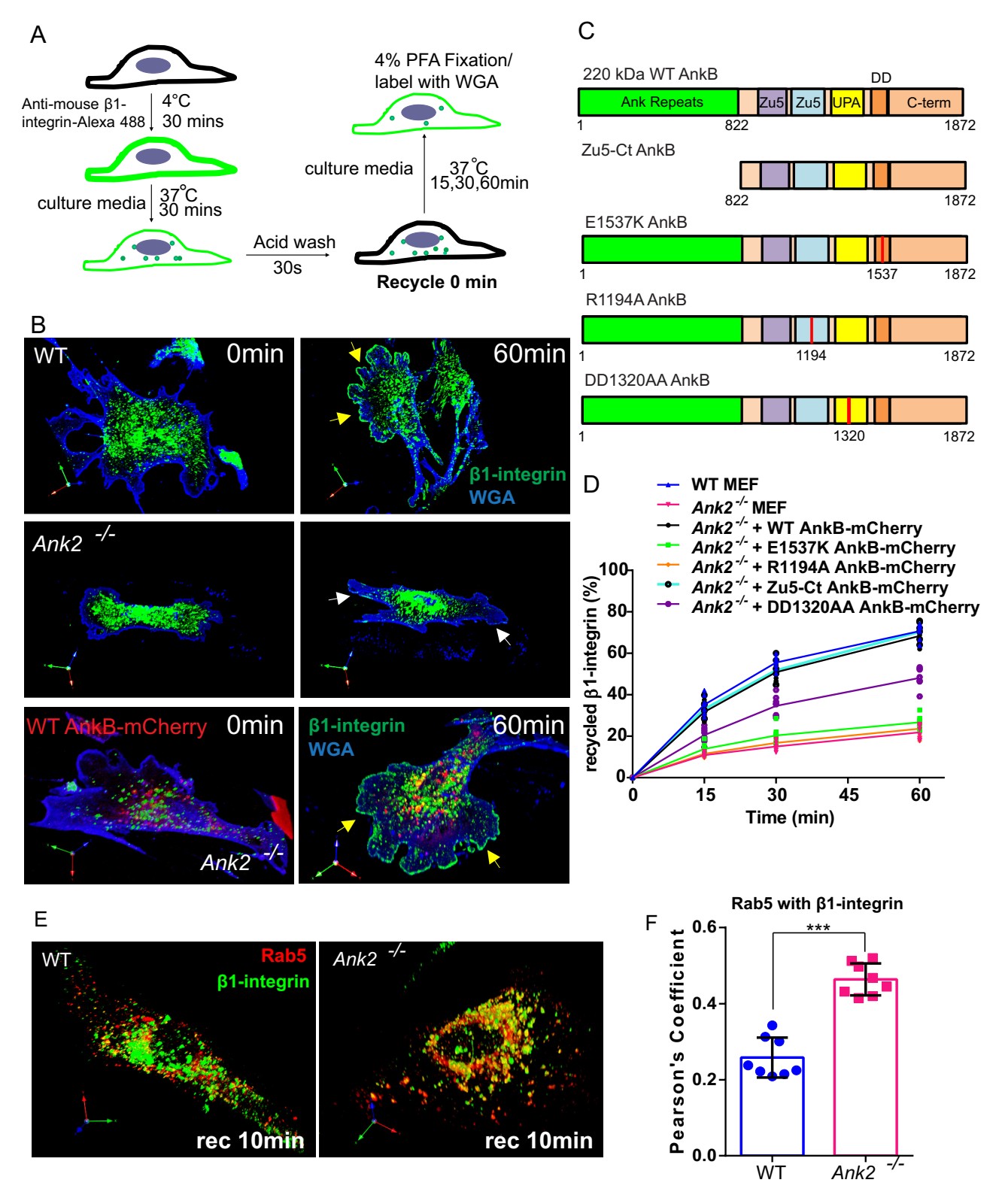

**Figure 6.** An AnkB-mediated mechanism promotes α5β1-integrin recycling to the plasma membrane of migrating MEFs. (**A**) Schematic representation of the β1-integrin recycling assay. Cell surface β1-integrins were labeled with anti-β1-integrin antibody conjugated with Alexa 488 at 4°C. Cells were incubated at 37°C for 30 min to allow internalization. Remaining cell surface labeling is reduced by acid wash and recorded as recycle time point 0 min. Cells were returned to 37°C incubation for indicated times following the wash with PBS and culture media. (**B**) Representative images of WT and *Ank2*[-/-]

*Figure 6 continued on next page*

Figure 6 continued

MEFs at recycling points t = 0 min and t = 60 min. β1-integrins are shown in green, plasma membranes labeled with WGA-Alexa 633 are shown in blue. Yellow arrows indicate plasma membrane areas at the migrating front containing recycled β1-integrin in WT and in *Ank2⁻/⁻* MEFs expressing WT AnkB-mCherry. White arrows indicate the absence of recycled β1-integrin signal at the plasma membrane of *Ank2⁻/⁻* MEFs (C) Schematic representation of the domain organization of AnkB-mCherry constructs used in structural function experiments. The various mutation sites are marked in red. (D) Quantitative data of β1-integrin recycling in WT, *Ank2⁻/⁻*, and *Ank2⁻/⁻* MEFs expressing WT or mutant AnkB-mCherry constructs. Data represents mean ± SD from five independent experiments. Individual data points indicate the mean from each experiment. N = 16. (E) Images show prolonged association of Rab5 with β1-integrin vesicles in *Ank2⁻/⁻* MEFs. (F) Pearson's co-localization coefficient between Rab5 and internalized β1-integrin 10 min post recycling in WT and *Ank2⁻/⁻* MEFs. Data represents mean ± SD for three independent experiments. ***p<0.001, two-tailed t-test. N = 8.

The following figure supplements are available for figure 6:

**Figure supplement 1.** Structural-functional study of AnkB-mCherry protein rescue of β1-integrin recycling deficits in *Ank2⁻/⁻* MEFs, β1-integrin localization to Rab22A- and Rab5-positive early endosomes.

**Figure supplement 2.** Knock down of RabGAP1L or replacement with GAP-deficient RabGAP1L affects cell viability.

**Figure supplement 3.** Overexpression of WT or constitutively active Rab22A impairs receptor recycling.

not of αvβ3-integrin, is consistent with reports that fibroblasts rely primarily on β1-integrins for establishing fibronectin haptotaxis (*King et al., 2016*).

Interestingly, rescue experiments in *Ank2⁻/⁻* MEFs expressing either WT or E1537K AnkB-GFP (lacking RabGAP1L binding) showed that expression of WT AnkB-GFP restored the haptotatic migration pattern in *Ank2⁻/⁻* MEFs while E1537K AnkB-GFP did not rescue the impaired haptotactic response (*Figure 7B,C*). Together, these data indicate that the AnkB-RabGAP1L interaction, which is required for polarized recycling of β1-integrin, is also required for efficient fibroblast migration along a fibronectin gradient. In future studies, it will be important to characterize the role of the AnkB-RabGAP1L pathway in cell migration in a more physiological relevant 3D micro-environment and to evaluate its role in cancer cell metastasis (*Caswell et al., 2007*, *2008*).

## Discussion

We report the discovery of a new pathway required for polarized membrane transport of specialized cargos. It was previously reported that AnkB promotes fast axonal transport through recruiting dynactin to PI3P-positive organelles (*Lorenzo et al., 2014*). We demonstrate that in fibroblasts AnkB functions as a PI3P effector associated with early endosomes, and describe a new interaction of AnkB with RabGAP1L. Moreover, we show that AnkB recruits RabGAP1L to PI3P-positive organelles, where it inactivates Rab22A, and identify α5β1-integrin as a specialized cargo that depends on AnkB/RabGAP1L for efficient recycling to the leading edge of migrating fibroblasts (*Figure 8*). Thus, our results establish AnkB as a key nodal element in the protein circuitry required for α5β1-integrin recycling and for directional fibroblast migration on fibronectin gradients.

An AnkB/RabGAP1L-based pathway with roles in polarized long-range organelle transport likely first appeared in jawed vertebrates over 400 million years ago as a result of neofunctionalization of duplicated genes. Human AnkB, RabGAP1L, and Rab22A all have homologues in ghost sharks and zebrafish with a high level of sequence similarity, including nearly complete conservation of their AnkB-RabGAP1L binding sites. In contrast, residues required for interaction between AnkB and RabGAP1L are highly divergent in the closest homologues of *Drosophila melanogaster* and *C. elegans*. Human AnkB and its paralogue AnkG exhibit over 70% sequence conservation, with only a few localized regions of divergence. One highly divergent site between these two ankyrins is an unstructured peptide connecting the MBD and the first ZU5 domain, which through intramolecular inhibition, prevents interactions between AnkB and membrane partners (*He et al., 2013*). Here, we identify an additional divergent site within AnkB's DD that allows AnkB, but not AnkG, to bind to RabGAP1L. AnkB, likely gained RabGAP1L-binding activity after divergence of AnkB and AnkG proteins in early vertebrate evolution, although, alternatively, AnkG may have lost this function. Similarly, RabGAP1L differs from its paralogue RabGAP1 primarily in the C-terminal residues required to bind AnkB.

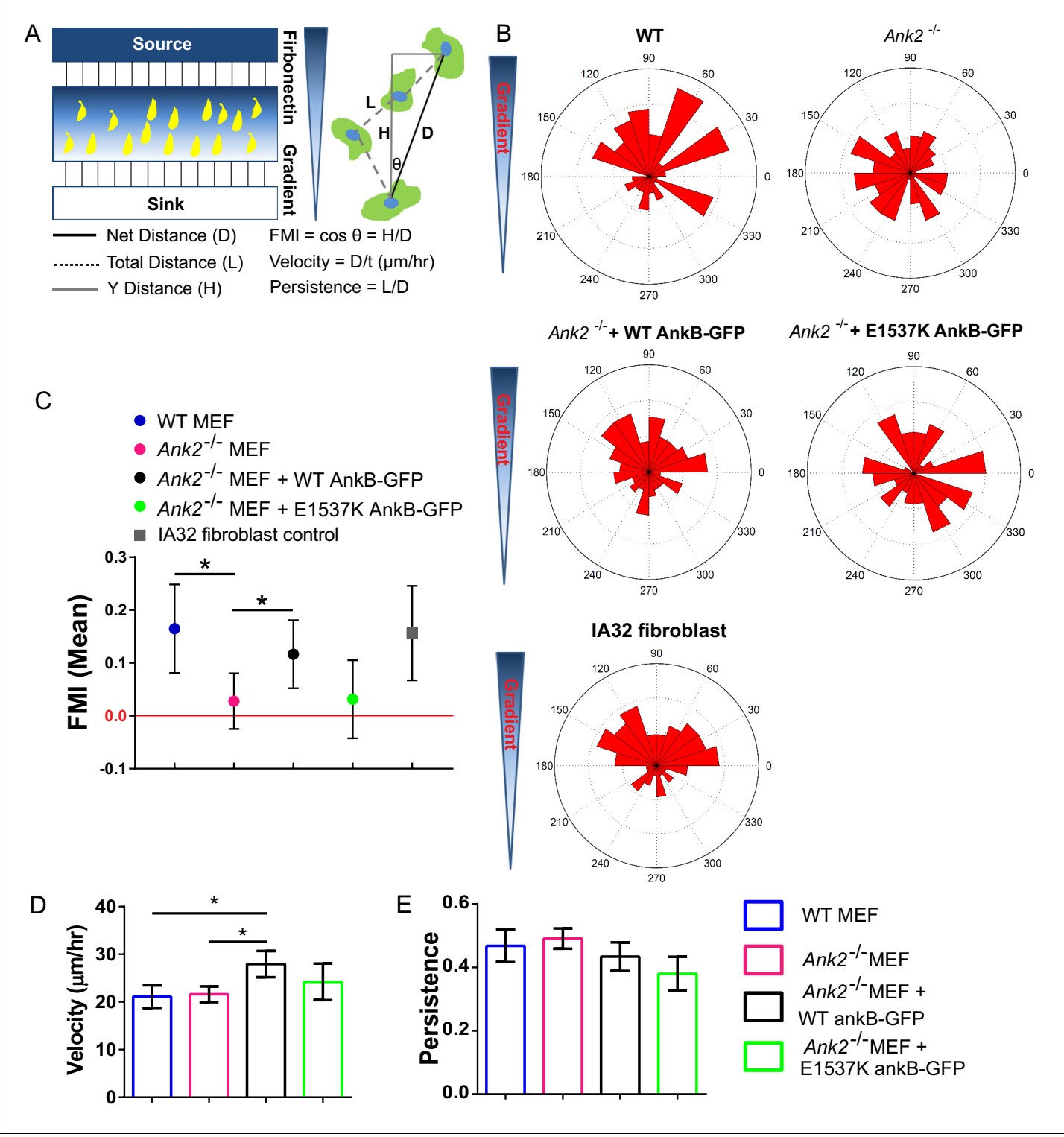

**Figure 7.** An AnkB-RabGAP1L complex is required for haptotaxis of MEFs on a fibronectin gradient. (**A**) Schematic of a microfluidic chamber system-based haptotaxis assay and analysis. (**B**) Rose plot showing the distribution of the tracking end point of cells migrate on a linear gradient of fibronectin. (**C**) Mean FMI of WT MEFs, control IA32 fibroblasts, *Ank2*$^{-/-}$ MEFs and *Ank2*$^{-/-}$ MEFs expressing WT AnkB-GFP or E1537K AnkB-GFP with 95% confidence interval (95% CI). Mean FMI with 95% CI crossing 0 is considered as not haptotaxing. (**D, E**) Mean velocity (**D**) and persistence (**E**) of the motility of WT, *Ank2*$^{-/-}$, and *Ank2*$^{-/-}$ MEFs expressing WT AnkB-GFP or E1537K AnkB-GFP. Data represent mean ± SD from four independent experiments. *p=0.0238, 0.04. N = 98, 156, 116, 70, 78. one-way ANOVA with Tukey post-test.

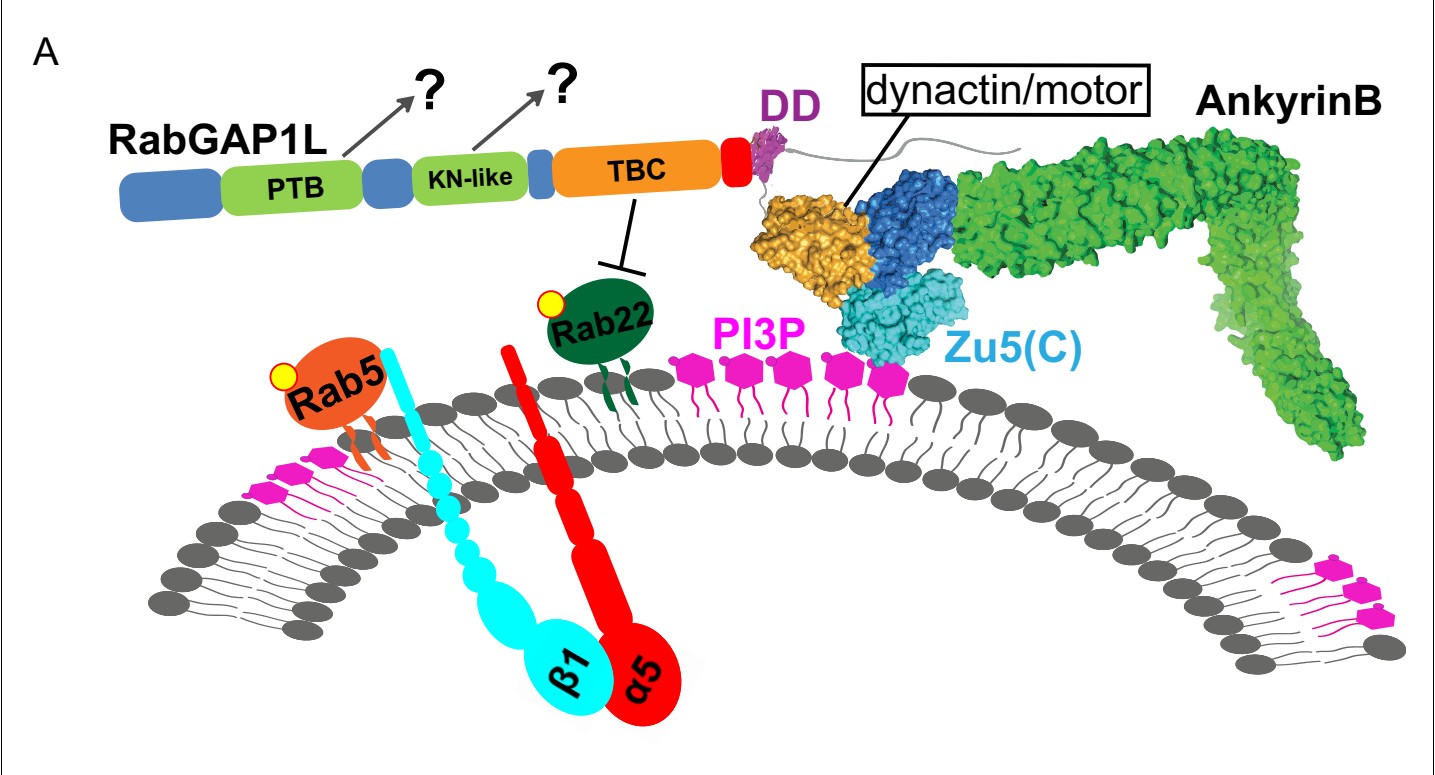

**Figure 8.** Model of AnkB-mediated mechanism for endosomal transport. (**A**) Model of AnkB-mediated mechanism for the recruitment of RabGAP1L to PI3P/Rab22A-positive endosomal compartments, which is critical for the maturation and recycling of α5β1-integrin containing endosomes.

The following figure supplement is available for figure 8:

**Figure supplement 1.** AnkB and RabGAP1L co-localize at the corpus callosum in the CNS and costameres in skeletal muscle.

A functional prediction, based on the recent evolution of the AnkB/RabGAP1L-based pathway, is that it will engage a subset of cargos with specialized roles in vertebrate physiology. In support of this idea, Rab22A, an AnkB/RabGAP1L substrate identified in this study, participates in recycling of other specialized cargos, including the Menkes copper transporter, the epidermal growth factor receptor (EGFR), and the major histocompatibility complex class I (*Holloway et al., 2013*; *Maldonado-Báez and Donaldson, 2013*; *Weigert et al., 2004*). Interestingly, Rab22A shares 70% sequence identity with Rab22B/Rab31, also implicated in the endosomal transport of specialized cargos such as EGFR and the p75 neurotrophin receptor (*Baeza-Raja et al., 2012*; *Chua and Tang, 2014*). It will be of interest to determine whether Rab22B is a RabGAP1L substrate, and whether it also depends of AnkB for its inactivation. Likewise, it will be important to identify the full complement of membrane-associated proteins engaged by the AnkB/RabGAP1L pathway.

As an initial step in further elucidating the physiological role(s) of the AnkB/RabGAP1L interaction, we performed a proximity ligation assay as well as immunofluorescent staining of total AnkB and RabGAP1L in brain and skeletal muscle tissues from PND 30 mice (*Figure 8—figure supplement 1*). We detected a strong PLA signal in the corpus callosum in the CNS and costameres in skeletal muscle (*Figure 8—figure supplement 1B,D*). Interestingly, AnkB is required for preservation of the corpus callosum and assembly of costameres (*Scotland et al., 1998*; *Lorenzo et al., 2014*; *Ayalon et al., 2008*). The cellular role of the AnkB/RabGAP1L interaction remains to be further characterized in more specialized mice models such as E1537K AnkB knock-in mice lacking RabGAP1L-binding activity.

An AnkB mutation eliminating interaction with the dynactin complex impairs but does not abolish α5β1-integrin recycling to the plasma membrane (*Figure 6D*). The residual α5β1-integrin transport may operate through alternative mechanisms to recruit motor proteins to AnkB-associated

organelles, and facilitate their traffic from the perinuclear compartment to the plasma membrane. One potential candidate is the kinesin-3 family member KIF16B, which binds directly to PI3P lipids and promotes outward transport of Rab5-associated early endosomes to the cell surface (*Hoepfner et al., 2005*).

Inactivation of Rab22A by RabGAP1L through its TBC domain likely contributes to the maturation of early endosomes. GTP-bound Rab22A directly associates with the Rab5 GEF Rabex-1, which activates Rab5 (*Zhu et al., 2009*). Alternatively, conversion of Rab22A to its GDP form through the AnkB-mediated recruitment of RabGAP1L would be expected to reduce Rab5 activation, thus promoting loss of early endosome identity. Another possibility is that RabGAP1L perform functions independent of GAP-activity, similar to p120RasGAP, which competes with, instead of inactivating, Rab21in binding to integrin α cytoplasmic tails to promote integrin recycling (*Mai et al., 2011*). However, whether and how loss of Rab22A association with early endosomes lead to the polarized transport of organelles preferentially to the front of migrating fibroblasts remains to be elucidated.

The regulation of GTPases and their adaptor proteins also involves kinases (*Stenmark, 2009*; *Frasa et al., 2012*). Specifically, previous studies had shown that Diacylglycerol kinase α (DGKα) is required for Rab-coupling protein (RCP)-dependent integrin trafficking and Akt mediated phosphorylation of ACAP1, a GAP for Afr6, is required for integrin recycling (*Rainero et al., 2012*; *Li et al., 2005*). Although the kinase(s) that regulate RabGAP1L activity has not been identified, it remains to be a possible regulatory mechanism for sensing directional cues during haptotaxis (*Figure 8*).

Integrin dynamics has been the focus of intense investigation with particular attention from investigators in the fields of cancer cell metastasis, and angiogenesis (*De Franceschi et al., 2015*; *Caswell et al., 2007*; *Dozynkiewicz et al., 2012*; *Paul et al., 2015*; *Tian et al., 2012*). Interestingly, Rab22A also contributes to exosome biogenesis and shedding from primary tumor cells and tumor metastasis (*Wang et al., 2014b*). Our discovery of an AnkB/RabGAP1L pathway as a key regulator of Rab22A localization and activity, and of α5β1-integrin polarized transport, offers new insights into the molecular circuitry underlying fibroblast and endothelial cell migration and tumor metastasis, as well as potential new targets for regulation.

Long-range transport and targeting of integrins underlies neurite outgrowth and neuronal migration (*Anton et al., 1999*; *Condic, 2001*; *Condic and Letourneau, 1997*; *Wu and Reddy, 2012*). Moreover, proper levels and dynamics of α5β1-integrin complexes, shown to localize to nerve growth cones, are required in multiple stages of brain development and synaptic function (*Bi et al., 2001*; *Graus-Porta et al., 2001*; *Marchetti et al., 2010*; *Yanagida et al., 1999*). Intriguingly, the neuron-specific 440 kDa AnkB isoform is exclusively targeted to axons and enriched at axonal growth cones and might play a role in axonal guidance (*Kunimoto, 1995*). It is, thus, conceivable that similar to its roles in migrating fibroblasts, an AnkB/RabGAP1L pathway might facilitate the 440 kDa AnkB-mediated axonal growth cone behavior, axonal guidance and synaptogenesis.

## Materials and methods

### Mouse lines and animal care

All animal care and procedures were approved by the Institutional Animal Care and Use Committee of Duke University. AnkB KO mice (*Scotland et al., 1998*) were generated by targeted disruption of the endogenous *Ank2* gene by homologous recombination. In brief, a clone containing 17 kb of the *Ank2* gene isolated from a 129SVJ genomic λ DNA library was modified to introduce a NotI site-flanked cassette containing a neomycin resistance gene, an in-frame HA epitope, and a stop codon within an exon in the spectrin-binding domain of AnkB. Male C57BL/6 chimeras containing the targeting construct were crossed to C57BL/6 females to assure germline transmission. Heterozygous carriers ($Ank2^{+/-}$) were maintained in a mixed 129SVJ/C57BL/6 genetic background and used to generate WT ($Ank2^{+/+}$) and AnkB KO mice ($Ank2^{-/-}$) littermates.

### DNA constructs

AnkB-2xHA, AnkB-mCherry, DD1320AA AnkB-mCherry, and R1194A AnkB-mCherry clones have been previously described (*Lorenzo et al., 2014*). Full-length EB1-GFP, LAMP1-GFP, Rab5-RFP, Rab11-GFP, and TGN38-GFP were purchased from Addgene (Addgene, Cambridge, MA). The pN1-DEST-mMaple3 vector used as backbone for the mMaple3-tagged clones was generated from the

pN1-DEST-mCherry vector. In brief, the AgeI-MfeI mCherry sequence was replaced with a 5'-AgeI-mMaple3-MfeI-3' fragment (cloned from Zyxin-mMaple3, a generous gift from Xiaowei Zhuang, Harvard University, MA). AnkB-mMaple3 and α5-integrin-mMaple3 clones were then generated by LR recombination (Invitrogen, Carlsbad, CA) of either AnkB-pENTER or α5-integrin-pENTER with pN1-DEST-mMaple3.

AnkB ZU5$^N$-ZU5$^C$-UPA-DD-Ct-mCherry was generated by replacing the GFP sequence from ZU5$^N$-ZU5$^C$-UPA-DD-Ct-GFP with the mCherry sequence using the PmeI and NotI sites. ΔDD AnkB-mCherry, E1537K AnkB-mCherry mutants were generated by site-directed mutagenesis. mMaple3-2×FYVE$^{EEA1}$ was generated by replacing the GFP sequence in the GFP-2×FYVE$^{EEA1}$ clone, a generous gift from P. De Camilli (Yale University, CT), with the mMaple3 sequence using the AgeI and XhoI sites.

The mouse RabGAP1L cDNA sequence was subcloned into pENTER using the D-TOPO approach. mCherry- and GFP-tagged RabGAP1L derivatives were generated by LR recombination. The mouse Rab22A cDNA was subcloned into the SalI and BamHI sites of pEGFP-C1 to generate GFP-Rab22A. Then, the GFP sequence was removed and replaced by a 5'-AgeI- mMaple3- AccIII-3'AgeI fragment. MBP-RabGAPL1(1–235)-6xHis was generated by LR recombination between RabGAP1 (1-235)-pENTER and the pMAL-c4G-DEST vector. Similarly, pGBKT7-AnkB DD and pGBKT7-AnkG DD clones were generated by LR recombination between AnkB DD-pENTER or AnkG DD-pENTER and the pGBKT7-DEST vector. pGADT7-RabGAP1L (E507-L815) were generated by LR recombination between RabGAP1L (E507-L815)-pENTER and pGADT7-DEST vector. All mutations were introduced by site-directed mutagenesis.

## Antibodies

An affinity-purified antibody against RabGAP1L was generated by immunization of rabbits with a purified peptide containing amino acids 1–235 of mouse RabGAP1L (see generation of anti-RabGAP1L antibody). Rabbit affinity-purified antibodies against total AnkB (C-terminal domain), β2-spectrin (spectrin repeats 4–9), GFP, sheep anti-AnkB (C-terminal domain) and goat anti-AnkG (C-terminal domain) were all generated in our laboratory (*Ayalon et al., 2011*). Other primary antibodies include, mouse anti-HA and chicken anti-GFP (Aves Labs, Tigard, OR), rabbit anti-Rab5 and anti-TGN38 (Cell Signaling Technology, Danvers, MA), mouse anti-LAMP1 (Developmental Studies Hybridoma Bank, University of Iowa, IA), rabbit anti-Rab11 (Invitrogen), rat anti-EEA1 and rabbit anti-golgin97 (Abcam, Cambridge, UK), mouse anti-α-tubulin and mouse anti-sheep/goat IgG (Thermo, Waltham, MA). Alexa488-anti-mouse/rat CD29 (Biolegend, San Diego, CA), Alexa633-WGA and Alexa568-tranferrin (Life Technologies, Carlsbad, CA) were used in internalization and recycling assays. Secondary antibodies used includes: Alexa Fluor 488 or 568–conjugated donkey anti–mouse, anti–rabbit, anti-chicken, anti-rat, anti-sheep or anti–goat purchased from Invitrogen.

## Generation of anti-RabGAP1L antibody

A His-MBP-RabGAP1L (residues 1–235) protein was purified from bacterial cultures by a two-step affinity purification protocol using nickel and amylose beads. MBP and His tags were removed by subsequent treatment with precision protease and affinity chromatography using GST beads. Purified RabGAP1L peptides were used as antigen for rabbit immunization. Serum from immunized rabbits was collected and the anti-RabGAP1L antibody purified using in-tandem ovalbumin-, MBP-, and antigen (RabGAP1L 1–235)-affinity columns. The eluted antibody was mixed 1:1 (volume) with glycerol and stored at −20°C.

## Mammalian cell culture and transfection

Primary fibroblasts were isolated from postnatal day zero (PND0) *Ank2$^{-/-}$* and WT pups and cultured by standard methods. Passage one or two MEFs were electroporated with plasmids using an ECM 830 Square Wave Electroporation System830 (BTX, Harvard Apparatus, Holliston, MA) following the standard protocol recommended by the manufacturer. Cells were either imaged live or fixed with 4% PFA for further examination.

## Live-cell imaging and analysis

Cells expressing fluorescently-tagged proteins were cultured on fibronectin-coated MatTek dishes and imaged using a Zeiss LSM 780 inverted confocal microscope equipped with temperature and $CO_2$ controls. Individual cells were selected for either one-frame or time-lapse imaging (1 frame/2 s, 60 frames). Tracks of individual vesicles were generated using the manual tracker function of ImageJ. Velocity (µm) = displacement/time and Persistence = displacement/total distance.

## Immunoprecipitation

Total protein homogenates from MEFs were prepared in PBS containing 150 mM NaCl, 0.32 M sucrose, 2 mM EDTA, 0.1% Triton X-100, 0.1% sodium deoxycholate, 0.1% SDS, and protease inhibitors (10 µg/ml AEBSF, 30 µg/ml benzamidine, 10 µg/ml pepstatin, and 10 µg/ml leupeptin; EMD Millipore, Billerica, MA). Samples were centrifuged at 100,000 g for 30 min, and the supernatants were precleared with protein A/G Dynabeads (EMD Millipore) and subjected to immunoprecipitation using antibodies against AnkB, AnkG, RabGAP1L, or control IgG. Immunoprecipitation samples were resolved by SDS-PAGE and Western blotting, and signal detected using the Odyssey CLx imaging system.

For coimmunoprecipitation experiments, $5 \times 10^6$ HEK293 cells were plated in 10 cm dishes and transfected with 4 µg of each plasmid using Lipofectamine 2000 (Invitrogen) according to the manufacturer's instructions. Cells were harvested 72 hr after transfection and lysed in 0.5% Triton X-100 in lysis buffer (10 mM sodium phosphate, 0.32 M sucrose, 2 mM EDTA, and protease inhibitors). Cell lysates were centrifuged at 100,000 g for 30 min, and the soluble fraction was collected and precleared by incubation with protein A/G Dynabeads. Coimmunoprecipitation experiments were performed using protein A/G Dynabeads and mouse anti-HA or rabbit anti-GFP antibodies. Immunoprecipitation samples were resolved by SDS-PAGE and Western blot as described in the previous paragraph.

## Yeast-two-hybrid assay

The Y2H screen was performed using the Matchmaker Gold Yeast-Two-Hybrid System and the Mouse Universal Normalized cDNA library (Clontech, Mountain View, CA). The screen was performed following the recommendations of the manufacturer. The GAL4 DNA-BD/bait construct was prepared by ligating a PCR fragment containing the AnkB death domain (aa1485-1563, accession no. NM_020977.3) into the pGBKT7 vector (Clontech). The yeast strain Y2HGold was maintained on YPD agar plates on SD –TRP (tryptophan) plates when transfected with the GAL4 DNA-BD/AnkB DD bait construct. To confirm the expression of the bait and prey plasmids, cells were grown on SD-Leu/Trp plates. The library was transformed into the AH 109 yeast strain (Takara Bio Inc., Mountain View, CA). The Y2H screen was performed under high-stringency growth conditions as recommended by the manufacturer.

Yeast cells coexpressing either AnkB DD or AnkG DD (baits) together with either the C-terminal sequence of RabGAP1L cloned in pGADT7, or the empty pGADT7vector (prey) were grown on plates lacking leucine, tryptophan, histidine, and adenine (-Leu, -Trp, -His, and -Ade) and supplemented with 125 ng/ml Aureobasidin A. Same conditions were used to assess interactions between AnkB DD mutant baits and the RabGAP1L C-terminal domain.

## Proximity ligation assay

PLA was performed using the commercial Duolink kit (Sigma-Aldrich, St. Louis, MO) following the manufacturer's recommendations. PFA-fixed cells or deparaffinized tissue sections were incubated overnight with a pair of primary antibodies, each produced in different species, against the putative interacting partners. Duolink minus- and plus-probes were used to detect antibody-labeled proteins.

## mMaple3-based photoconversion assay and analysis

Cells were transfected with plasmids encoding WT or E1537K AnkB-mMaple3 or α5-integrin-mMaple3 and plated on fibronectin-coated MatTek dishes with a silicone insert. Inserts were removed 48 hr post-plating to allow cell migration towards the exposed region. Migrating cells at the edges of the dish were selected for photoconversion and imaged by time-lapse video microscopy. In brief, the perinuclear region of mMaple3-expressing cells was stimulated with blue light (λ = 405 nm) to

convert mMaple3 particles from green to red fluorescence. Photoconverted cells were continuously imaged for 16 min (1frame/15 s, 64 frames). The fluorescent intensity (FI) of red and green particles within the perinuclear region was determined using the Volocity software (Perkin Elmers, Waltham, MA). Retrograde transport of non-converted mMaple3-tagged vesicles towards the perinuclear region, expressed as percentage of green fluorescent gain, was calculated as: [FI at 16 min – FI at 45 s (post-conversion time))/ (FI at 0 s – intensity at 45 s)]. The anterograde transport of mMaple3-tagged vesicles was expressed as percentage of red fluorescent loss and calculated as [(intensity at 45 s– intensity at 16 min)/ (intensity at 45 s – intensity at 0 s)]. The transport of converted mMaple3 vesicles from the perinuclear region is expressed as the gain of fluorescent intensity at the front or rear cell ends at each time point.

## Immunolabeling
Cells were washed with room temperature PBS, fixed for 15 min at room temperature in a 4% para-formaldehyde solution in PBS, and permeabilized with 0.2% vol/vol Triton X-100 in PBS for 10 min. Samples were then blocked for 60 min in blocking buffer (3% BSA, 0.2% Tween-20 in PBS) and incubated overnight with primary antibodies in blocking buffer at 4°C. Cells were washed three times with PBS, incubated with fluorescent-labeled secondary antibody conjugates in blocking buffer at room temperature, washed three times with PBS, and mounted in Pro-Long Gold mounting media (Life Technologies, Carlsbad, CA).

## Cell surface protein biotinylation and recycling assay
Cells were rinsed twice with ice cold PBS and labeled with 0.2 mg/ml sulfo-NHS-SS biotin (Thermo) in PBS at 4°C for 30 min. The remaining sulfo-NHS-SS biotin was quenched with 50 mM Tris pH 8.0 in PBS, and the cells were washed two more times with PBS. To allow endocytosis, pre-warmed medium was added to the cells and the cultures were incubated at 37°C for various time points before fixation with 4% PFA. After 30 min of endocytosis, remaining surface exposed biotin labels were removed by incubating the cells 2 × 5 min in cold 100 mM MESNa buffer (50 mM Tris, 100 mM NaCl, 1 mM EDTA, 0.2 wt/vol BSA, pH 8.6). Cells were returned to a 37°C incubator to allow the recycling of biotin-labeled internalized proteins. Cells were fixed at various time points and the biotin-labeled proteins detected by Streptavidin-Alexa 488 (Life Technologies).

## Integrin and transferrin recycling assay
Cells were incubated with either Alexa 488-anti-mouse/rat β1-integrin, Alexa 488-anti-mouse/rat β3-integrin (BioLegend), or Alexa 568-transferrin (Life Technologies) in DMEM containing 0.5% FBS (DMEM-FBS) at 4°C for 30 min followed by washes with ice-cold PBS and DMEM-FBS. To allow endocytosis, fluorescently-labeled cultures were incubated at 37°C for 30 min. The remaining surface-associated fluorescence was quenched by brief acid wash (0.5% acetic acid, 0.5 M NaCl, pH 3.0). Following internalization cells were washed with PBS and, either fixed (recycling time 0 min), or incubated at 37°C for the indicated times before fixation (recycling time 15 min, 30 min, 60 min). The percentage of recycled proteins at a given time point (t) is expressed as: fluorescent intensity on the plasma membrane at time point (t)/fluorescent intensity in cytoplasm at time point (0). The percentage of remaining intracellular transferrin at a given time point is expressed as a ratio of the fluorescent intensity remaining in the cytoplasm at time point (t)/fluorescent intensity in cytoplasm at time point (0).

## shRNA mediated knock down of RabGAP1L and viability test
The PLKO-BFP-Tet-on vector was generated as described in *He et al. (2013)*. The production of lentivirus with vectors inserted with the shRNA hairpin targeting mouse RabGAP1L and the subsequent infection of cultured cells were performed following a standard protocal (*He et al., 2013*). Hairpins used were: luciferase control (5′-GGAGATCGAATCTTAATGTGC-3′) and mouse RabGAP1L (5′-TGGAACAGGCTTGCAATATT -3′).

2 × 10⁴ cells were plated in matak plate and treated with 4 µg/ml doxycycline the next day (Day1). Cells were transfected with RabGAP1L rescue plasmids 8 hr later and the number of cells were counted every 24 hr.

## Haptotaxis assay and analysis

The haptotaxis assay was performed as previously described (*Wu et al., 2012*). WT or *Ank2$^{-/-}$* MEFs were plated on microfluidic chambers containing a linear fibronectin gradient. *Ank2$^{-/-}$* MEFs transfected with WT AnkB-GFP or E1537K AnkB-GFP were sorted, and GFP-positive cells were pooled and plated on microfluidic chambers as described above. Cells were allowed to migrate for 24 hr on the chamber, and then were manually tracked allowing the calculation of FMI, velocity and persistence of migration, as described in *Figure 7A*. Rose plots of directional migration on normalized polar coordinates were generated using a MATLAB script described in *King et al., 2016*.

## Replication and statistical analysis

Each experiment was repeated using at least three lines of mouse embryonic fibroblast isolated from three individual PND0 WT (*Ank2$^{+/+}$*) and AnkB KO (*Ank2$^{-/-}$*) littermates, which is considered as biological replicates. Each experiment was done three or more times, which is considered as technical replicates. All of the data were combined for statistical analysis unless otherwise stated.

GraphPad Prism (GraphPad Software, La Jolla, CA) was used for statistical analysis. Statistical differences were determined by unpaired Student's *t*-test or by repeated measures one-way ANOVA, followed by a post hoc Tukey's test. Results are presented as mean ± SD. Significance was considered as p≤0.05. Exact p-values were shown when p>0.001.

## Acknowledgements

This work was supported by the Howard Hughes Medical Institute to V Bennett and by National Institute of Health grant GM110155 to JE Bear.

## Additional information

### Funding

| Funder | Grant reference number | Author |
| --- | --- | --- |
| Howard Hughes Medical Institute | | Fangfei Qu<br>Damaris N Lorenzo<br>Vann Bennett |
| National Institutes of Health | GM110155 | Samantha J King<br>Rebecca Brooks<br>James E Bear |

The funders had no role in study design, data collection and interpretation, or the decision to submit the work for publication.

### Author contributions

FQ, Conception and design, Acquisition of data, Analysis and interpretation of data, Drafting or revising the article; DNL, Conception and design, Analysis and interpretation of data, Drafting or revising the article, Contributed unpublished essential data or reagents; SJK, RB, Acquisition of data, Analysis and interpretation of data, Drafting or revising the article; JEB, VB, Conception and design, Drafting or revising the article

### Author ORCIDs

Vann Bennett, http://orcid.org/0000-0003-2695-7209

### Ethics

Animal experimentation: This study was performed strictly following the guide for the laboratory animal care and use at Duke University Medical Center. All of the animals were handled according to approved Institutional Animal Care and Use Committee (IACUC) protocol (# A149-15-05) of Duke University.

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
