## [Decision Letter]

Thank you for submitting your article "Ankyrin-B is a PI3P effector that promotes polarized α5β1-integrin recycling via recruiting RabGAP1L to early endosomes" for consideration by *eLife*. Your article has been favorably evaluated by Anna Akhmanova as the Senior Editor, Johanna Ivaska as the Reviewing Editor, and three reviewers including Jim Norman (Reviewer #3).

The reviewers have discussed the reviews with one another and the Reviewing Editor has drafted this decision to help you prepare a revised submission. All the reviewers found the study largely convincing and interesting. Therefore, we would like to invite resubmission of a satisfactorily revised manuscript.

Summary:

In this manuscript, the authors demonstrated a role of Ankyrin-B (AnkB) with its binding partner RabGAP1L in polarized trafficking of α5β1 integrin. They first showed that AnkB mediates not only axonal trafficking in neurons as reported previously but also long-range transport of PI3P-containing organelles such as early endosomes and lysosomes in non-neuronal cells. In migrating fibroblasts, α5β1 integrin was found to be one of the important cargos of this trafficking pathway and was recycled in a polarized manner during persistent cell migration. To investigate the mechanism of how AnkB functions, the authors identified RabGAP1L, a GAP for Rab22A, as a novel binding partner of AnkB. From structural and evolutionary perspectives, they carefully determined critical residues for the interaction between AnkB and RabGAP1L. Finally, they showed that the interactions of AnkB with RabGAP1L, PI3P, and in part dynactin are required for polarized recycling of α5β1 integrin in migrating fibroblasts by a yet undefined mechanisms of cargo recruitment. There are also some data suggesting that AnkB may contribute to haptotaxis toward the α5β1 integrin ligand, fibronectin.

Overall, the study is mostly well-designed and high quality, and these findings advance our knowledge of how a scaffolding protein organizes Rab GTPases, phosphoinositides, and microtubule motors during polarized endosomal trafficking.

Essential revisions:

1) The rationale behind the choice of the model system (MEFs) is unclear limiting the biological relevance of these findings. In the first paragraph of the subsection “AnkB promotes long-range transport of PI3P-positive organelles” the authors mention that they wanted to investigate whether AnkB contributes to long-range transport of vesicles also in cell types other than neurons. They proceed by jumping into overexpression of AnkB in MEFs. Why MEFs? What is the biological relevance of long-range transport in these cells? Which tissues/organs, other than the CNS, express AnkB? What is the evidence from the *AnkB^-/-^* mice that fibroblast function would be compromised in the absence of AnkB? The MEFs are obviously a convenient model system for structure-function cell biological experimentation but a clear indication of the biological relevance of this pathway would be important to warrant publication in a journal like *eLife*. It would seem more relevant to include neurons in this study, also given the expertise of the Bennett laboratory and show the effect of the ANKB-E1537K mutant (RabGAP1L-binding-deficient) on axonal transport in neurons.

2) Data in Figure 7 are currently not convincing as the MEFs are hardly haptotaxing (an FMI of 0.1 is very low and hardly significantly different from 0.0). The authors should present spider/track plots and possibly vector plots rather than just this FMI metric, which is not really meaningful on its own as well as some movies of the cells in the FN gradient. Alternatively, the authors are encouraged to identify another cell line with stronger haptotaxis or to explore possibilities of looking at migration in 3D microenvironments.

3) What is lacking in the present study is the direct evidence that Rab22A is responsible for trafficking of α5β1 integrin downstream of the AnkB-RabGAP1L pathway. Does knockdown of RabGAP1L or overexpression of a constitutively active mutant of Rab22A phenocopy the deficiency in *AnkB^-/-^* cells?

4) A rescue experiment using WT and a GAP activity-deficient mutant of RabGAP1L is highly recommended to determine whether RabGAP1L-mediated Rab22A inactivation is actually involved in persistent cell migration.

5) Currently, imaging of endogenous AnkB and RabGAP1L are lacking. Given the functionality of the antibodies in immunofluorescence based assays (PLA (Figure 2)) the authors need to provide staining of the endogenous to validate their observations obtained with the over-expressed tagged constructs.

6) The Introduction and Discussion of the paper are inadequately referenced in regard of the existing literature on α5 integrin trafficking. Two labs in particular (Ivaska and Norman) have published numerous papers over of the last 15 years on this process, which elucidate a fair degree of the machinery responsible for controlling the endosomal trafficking of α5β1, and many other workers have complimented this and added other components to the machinery. For instance, Rab21, p120RasGAP (Ivaska), RCP (Norman, Ivaska), DGKα (Norman), ACAP1 (Hsu), WASH (Machesky/Norman), SNX17 (Cullen, Fassler) and EHD1 (Caplan) are known regulators of α5β1-integrin endosomal recycling, and Rab25 (Norman) and CLIC3 (Norman) control recycling of active α5β1-integrin from late endosomes/lysosomes. This fairly extensive body of literature should be adequately cited and discussed.

7) The authors should define clearly what they mean by polarised delivery of α5β1 to the cell front. If one demonstrates (as they have) a greater number of vesicles leaving the front portion of the perinuclear region and proceeding toward the cell front than to the back, this is good, but it's not evidence to support Bretscher's postulate that integrins are disengaged from the substratum and the cell rear and then moved to the front in vesicles. As the paper currently stands, it reads a little like their evidence does support such a process.

Also, they need to mention that there are papers demonstrating spatial restriction of α5β1 recycling at the cell front (Norman) and 'polarised delivery' specifically to the cell rear (Norman, Straube). All of these things are likely going on in various proportions at the same time, so the situation is far more complex than Mark Bretscher envisaged and this need careful discussion.

---

## [Author Response]

*[…] Essential revisions:*

*1) The rationale behind the choice of the model system (MEFs) is unclear limiting the biological relevance of these findings. In the first paragraph of the subsection “AnkB promotes long-range transport of PI3P-positive organelles” the authors mention that they wanted to investigate whether AnkB contributes to long-range transport of vesicles also in cell types other than neurons. They proceed by jumping into overexpression of AnkB in MEFs.*

All experiments with the expression of fluorescently tagged ankyrin-B were done in primary MEFs isolated from ankyrin-B null (*Ank2 ^-/-^*) mice that have complete depletion of 220kDa ankyrin-B (as shown in Figure 1). We were particularly careful not to overexpress ankyrin-B since this led to cell death and mentioned this in the first paragraph of the subsection “AnkB promotes long-range transport of PI3P-positive organelles”. To make it clear to readers, we stated the genotype of MEFs we have used in every figure legend.

*Why MEFs? What is the biological relevance of long-range transport in these cells?*

As noted at the beginning of the Results, we wanted to extend the novel finding of a role of ankB in organelle transport in neurons to another cell type. We selected MEFs because these cells are a standard laboratory model that have been extensively characterized with respect to organelle transport. Moreover, we are using primary MEFs isolated from PND 0 neonates without any transformation or extensive passage. In addition, ankyrin-B null MEFs directly isolated from *Ank2^-/-^*mice can be rescued with wild-type levels of ankyrin-B, making it a perfect system to dissect out the molecular mechanism underlying the ankB-mediated organelle transport.

We have added a sentence describing the rationale behind using MEFs to study the function of ankyrin-B in organelle transport at the beginning of the Results section.

*Which tissues/organs, other than the CNS, express AnkB? What is the evidence from the AnkB^-/-^ mice that fibroblast function would be compromised in the absence of AnkB? The MEFs are obviously a convenient model system for structure-function cell biological experimentation but a clear indication of the biological relevance of this pathway would be important to warrant publication in a journal like eLife.*

AnkB is broadly expressed in multiple tissues other than CNS, including heart, skeletal muscle, pancreas and fat tissue, liver and skin. It is difficult to assign a specific physiological role of ankyrin-B in MEFs or define an exact phenotype as a direct evidence for compromised fibroblast function since ankyrin-B null (*Ank2 ^/-^*) mice die soon after birth with multisystem abnormalities including cardiac arrhythmia, severe CNS abnormalities, congenital myopathy, and pancreatic insufficiency (Scotland et al., 1998; Tuvia et al., 1999; Mohler et al., 2003; Ayalon et al., 2008; Healy et al., 2009; Lorenzo et al., 2014; Lorenzo et al., 2015). We include new data based on the proximity ligation assay demonstrating a nanoscale ankyrin-B-RabGAP1L interaction in commissural axons in the CNS and costameres in skeletal muscle as well as the immunofluorescence staining of total AnkB and RabGAP1L in these tissues in Figure 8—figure supplement 1 and mention the result in the fourth paragraph of the Discussion. To address the biological significance of these interactions, an E1537K AnkB knock-in mouse would be an ideal model to study. Unfortunately, the development of a knock-in mouse takes at least a year and, in our opinion, is beyond the scope of this initial discovery paper.

*It would seem more relevant to include neurons in this study, also given the expertise of the Bennett laboratory and show the effect of the ANKB-E1537K mutant (RabGAP1L-binding-deficient) on axonal transport in neurons.*

As described above, we did not use neurons because our purpose was to determine if cells other than neurons utilized ankyrin-B in organelle transport.

However, a potential role of neuronal specific isoform of ankyrin-B (440 kDa AnkB) has been suggested in axonal growth cones and axonal guidance been implicated (Kunimoto, 1995). While it is definitely worth investigating the phenotype of growth cones baring E1537K 440 kDa AnkB and the role of 440 kDa AnkB-RabGAP1L interaction in axonal guidance and synaptogenesis, we believe this is beyond the scope of this current paper, but worth future studies in our lab. This is mentioned in the last paragraph of the Discussion.

2) Data in Figure 7 are currently not convincing as the MEFs are hardly haptotaxing (an FMI of 0.1 is very low and hardly significantly different from 0.0).

Our key determinant of whether cells are haptotaxing is whether the 95% confidence intervals of the forward migration index (FMI) encompass 0. The FMI metric has been used in many previous publications (King et al. 2016, Chan et al. 2014, Asoken et al. 2014) to determine haptotactic fidelity. Although a value of 0.1-0.2 is low when compared to those often obtained for chemotaxis of cells such as neutrophils, the value is similar to FMI values for fibroblast haptotaxis that have previously been published (King et al. 2016, Chan et al. 2014), and is in the range expected for a process that is a response to short-range insoluble cues. In response to the reviewers’ concerns regarding the FMI values we have included the FMI value obtained for IA32 fibroblasts run in the same experiments as our MEFs in Figure 7. These are a cell type that we have previously published haptotaxis FMIs with, to show that this value is similar to the WT MEFs. It is also mentioned in the first paragraph of the subsection “AnkB/RabGAP1L interaction promotes directional cell migration”. Crucially, the FMI values obtained for the WT MEFs and the *Ank2^-/-^* MEFs rescued by WT-ankB-GFP are significantly different from those obtained for *a^-/-^*MEFs and the *AnkB^-/-^* MEFs rescued by E1537K-ankB-GFP, demonstrating an effect on haptotaxis.

*The authors should present spider/track plots and possibly vector plots rather than just this FMI metric, which is not really meaningful on its own as well as some movies of the cells in the FN gradient.*

We agree that a visual representation of haptotaxis with rose plots is a good idea and we have now included rose plots, which have been published in various directional migration studies and should give a clear visual representation of the migration. They are now shown in Figure 7. We understand the reviewers’ request for the inclusion of example movies of the cells in the FN gradient. However, unfortunately unlike other forms of directed migration such as chemotaxis, haptotaxis is extremely difficult to see by eye in the movies (hence the aforementioned lower FMI values) with analysis of the tracks being absolutely required to ascertain whether the cells are haptotaxing or not. Additionally, there are only a few cells per movie. Therefore, we respectfully feel that inclusion of these movies would not help with the readers understanding in this case, and rather would detract from the purpose of their inclusion, and so have not included them here.

*Alternatively, the authors are encouraged to identify another cell line with stronger haptotaxis or to explore possibilities of looking at migration in 3D microenvironments.*

We have addressed the reviewers’ concerns regarding our MEFs haptotacic fidelity by including the FMI values of a previously published cell line (IA32 fibroblasts) that was run simultaneously with our MEFs. These cells show a similar FMI value to our WT MEFs, as is shown in Figure 7, demonstrating that the FMI values seen are not low for a haptotaxis measurement in fibroblasts. We acknowledge that it would be informative to evaluate the role of AnkB-RabGAP1L pathway in cell migration in a more physiological relevant 3D micro-environment and its role in metastasis, and have mentioned this point in the Results section, last paragraph. However, we think these experiments are beyond the scope of this initial discovery paper.

*3) What is lacking in the present study is the direct evidence that Rab22A is responsible for trafficking of α5β1 integrin downstream of the AnkB-RabGAP1L pathway. Does knockdown of RabGAP1L or overexpression of a constitutively active mutant of Rab22A phenocopy the deficiency in AnkB^-/-^ cells?*

To address the requirement of RabGAP1L in α5β1-integrin trafficking, we generated MEFs with shRNA-mediated knockdown of RabGAP1L. Cells with ~70% knockdown of RabGAP1L exhibited multiple deficits including altered morphology and cell death. Therefore, it would be difficult to perform as well as hard to interpret the endosomal trafficking or integrin recycling experiment in RabGAP1L knockdown cells. We mention this result in a new subsection: "Balanced Rab22A activity is critical for β1-integrin recycling", and Figure 6—figure supplement 2 now summarizes the characterization of the growth and survival rate of RabGAP1L knock-down cells.

To address the concerns regarding the lack of direct evidence that Rab22A is responsible for trafficking of α5β1-integrin downstream of the AnkB-RabGAP1L pathway, we performed β1-integrin recycling assay in MEFs overexpressing WT Rab22A or constitutively active Rab22A. WT MEFs over-expressing WT Rab22A showed a reduced rate of β1-integrin recycling. The deficit was more severe in MEFs expressing constitutively active Q64L Rab22A, suggesting that the balanced Rab22A activity is critical for β1-integrin recycling. The result is summarized in Figure 6—figure supplement 3 and mentioned in a new subsection: "Balanced Rab22A activity is critical for β1-integrin recycling".

*4) A rescue experiment using WT and a GAP activity-deficient mutant of RabGAP1L is highly recommended to determine whether RabGAP1L-mediated Rab22A inactivation is actually involved in persistent cell migration.*

As we described above, shRNA-mediated knock down of RabGAP1L is lethal. Expression of shRNA-resistant WT RabGAP1L simultaneously after activation of the shRNA system rescued the lethal phenotype. However, expression of GAP-deficient RabGAP1L (R584A RabGAP1L) did not restore viability. It is therefore difficult to interpret cell migration because these cells are likely in the process of dying. This result suggests that RabGAP1L plays critical roles in cellular events in addition to polarized transport of α5β1-integrin and cell migration, which itself is an interesting question worth investigating in future studies. Therefore, we have summarized these results in Figure 6—figure supplement 2 and stated this finding in the first paragraph of the subsection “Balanced Rab22A activity is critical for β1-integrin recycling”.

*5) Currently, imaging of endogenous AnkB and RabGAP1L are lacking. Given the functionality of the antibodies in immunofluorescence based assays (PLA (Figure 2)) the authors need to provide staining of the endogenous to validate their observations obtained with the over-expressed tagged constructs.*

We examined the distribution of endogenous RabGAP1L as well as ankyrin-B using the same antibodies employed in the proximity ligation assay. We have confirmed the co-localization of Ankyin-B and RabGAP1L by immunofluorescence staining of endogenous proteins and added this result in Figure 2—figure supplement 1. The result is stated in the fifth paragraph of the subsection “AnkB directly recruits RabGAP1L to PI3P-positive organelles” as well. Figure 3—figure supplement 1 now summarizes our analysis of endogenous RabGAP1L localization to PI3P-positive organelles and Rab22A-positive organelles in WT and AnkB ^-/-^ MEFs and these results are mentioned in the last paragraph of the subsection “AnkB directly recruits RabGAP1L to PI3P-positive organelles” and in the first paragraph of the subsection “AnkB promotes dissociation of Rab22A from PI3P-associated organelles through recruitment of RabGAP1L”.

*6) The Introduction and Discussion of the paper are inadequately referenced in regard of the existing literature on α5 integrin trafficking. Two labs in particular (Ivaska and Norman) have published numerous papers over of the last 15 years on this process, which elucidate a fair degree of the machinery responsible for controlling the endosomal trafficking of α5β1, and many other workers have complimented this and added other components to the machinery. For instance, Rab21, p120RasGAP (Ivaska), RCP (Norman, Ivaska), DGKα (Norman), ACAP1 (Hsu), WASH (Machesky/Norman), SNX17 (Cullen, Fassler) and EHD1 (Caplan) are known regulators of α5β1-integrin endosomal recycling, and Rab25 (Norman) and CLIC3 (Norman) control recycling of active α5β1-integrin from late endosomes/lysosomes. This fairly extensive body of literature should be adequately cited and discussed.*

We thank reviewers for this detailed reminder about the existing literature dealing with α5-integrin trafficking. In the Introduction, we mainly focused on general issues related to endosomal trafficking and did not discuss specific cargo. The first mention of integrin trafficking occurred in the Results section: "Identification of α5β1-integrin trafficking as an AnkB-dependent cargo". We have attempted to properly cite appropriate papers in Results and Discussion sections in this revised manuscript.

*7) The authors should define clearly what they mean by polarised delivery of α5β1 to the cell front. If one demonstrates (as they have) a greater number of vesicles leaving the front portion of the perinuclear region and proceeding toward the cell front than to the back, this is good, but it's not evidence to support Bretscher's postulate that integrins are disengaged from the substratum and the cell rear and then moved to the front in vesicles. As the paper currently stands, it reads a little like their evidence does support such a process.*

*Also, they need to mention that there are papers demonstrating spatial restriction of α5β1 recycling at the cell front (Norman) and 'polarised delivery' specifically to the cell rear (Norman, Straube). All of these things are likely going on in various proportions at the same time, so the situation is far more complex than Mark Bretscher envisaged and this need careful discussion.*

We regret confusion caused by term "polarized delivery of α5-integrin". To clarify, we are specifically looking at the transport of perinuclear α5-integrins either to the front or rear of migrating cells. We do not intend to make any claim in regards to the origin of these integrins, regardless of whether they are internalized from the cell rear or the cell front. To make it clear to readers, we specifically changed "polarized transport/delivery of α5-integrins" to "polarized transport/delivery of perinuclear α5-integrins" in the text. Regarding a detailed discussion of Mark Bretscher’s model, we make a single reference to his 1989 and 1992 papers, which were written before contributions of Rabs were understood. We changed "polarized delivery" to "polarized delivery from cytoplasmic compartments to the leading edge of migrating fibroblasts" in the subsection “Identification of α5β1-integrin as an AnkB/RabGAP1L-dependent cargo”, where we cited his papers. His finding (Bretscher, 1989) actually stated internalization occurred all over the cell and did not specify a particular location: "[…] This directed flow is produced by the endocytosis of lipids plus receptors randomly over the cell surface combined with their return to the cell surface at the leading edge […] the fibronectin receptor follows a similar course to other circulating receptors". We acknowledge that recent studies utilizing the photoactivation strategy (Zech et al., 2011; Dozynkiewicz et al., 2012) revealed that the situation of integrin trafficking/recycling is much more complex than the initial Brescher's model and feel that a detailed discussion of all of the subsequent papers on this topic is more appropriate for a review focused on integrins, which we cite (Caswell et al., 2009; Bridgewater et al., 2012; Arjonen et al., 2012; De Franceschi et al., 2015).